behaviour, ecology

population viability analysis, demographic variance, vital rates, reproductive skew, poaching, extinction risk

**Author for correspondence:**
Nick Harvey Sky
e-mail: nick.c.harvey@gmail.com

# Female reproductive skew exacerbates the extinction risk from poaching in the eastern black rhino

Nick Harvey Sky[1,2], John Jackson[3], Geoffrey Chege[4], Jamie Gaymer[5], David Kimiti[6], Samuel Mutisya[7], Simon Nakito[7] and Susanne Shultz[1]

[1]Department of Earth and Environmental Sciences, University of Manchester, Manchester M13 9PL, UK
[2]North of England Zoological Society, Chester Zoo, Caughall Road, Chester CH2 1LH, UK
[3]Department of Zoology, University of Oxford, Oxford OX1 3SZ, UK
[4]Lewa Wildlife Conservancy, PO Box, Private Bag, Isiolo 60300, Kenya
[5]Ol Jogi Ltd., PO Box 259-10400, Nanyuki, Kenya
[6]Grevy's Zebra Trust, PO Box 15351-00509, Nairobi, Kenya
[7]Ol Pejeta Conservancy, PO Box 167, Nanyuki, Kenya

NH, 0000-0001-7824-0045; JJ, 0000-0002-4563-2840; SS, 0000-0002-7135-4880

Variation in individual demographic rates can have large consequences for populations. Female reproductive skew is an example of structured demographic heterogeneity where females have intrinsic qualities that make them more or less likely to breed. The consequences of reproductive skew for population dynamics are poorly understood in non-cooperatively breeding mammals, especially when coupled with other drivers such as poaching. We address this knowledge gap with population viability analyses using an age-specific, female-only, individual-based, stochastic population model built with long-term data for three Kenyan populations of the Critically Endangered eastern black rhino (*Diceros bicornis michaeli*). There was substantial reproductive skew, with a high proportion of females not breeding or doing so at very low rates. This had a large impact on the projected population growth rate for the smaller population on Ol Jogi. Moreover, including female reproductive skew exacerbates the effects of poaching, increasing the probability of extinction by approximately 70% under a simulated poaching pressure of 5% offtake per year. Tackling the effects of reproductive skew depends on whether it is mediated by habitat or social factors, with potential strategies including habitat and biological management respectively. Investigating and tackling reproductive skew in other species requires long-term, individual-level data collection.

## 1. Introduction

With over 1 million species facing extinction [1] and a 68% average decline in monitored vertebrate populations between 1970–2016 [2] quantifying extinction risk and identifying its drivers in vulnerable populations is crucial for biodiversity conservation. Small populations are particularly vulnerable to extinction [3] because of environmental and demographic stochasticity [4], inbreeding depression, loss of genetic diversity [5] and inverse density dependence (or Allee effects) [6], which all become more important as population size decreases. Understanding the drivers of extinction and population dynamics in small populations is vital to the development of effective conservation strategies.

The causes of unequal realized vital rates (survival, growth and reproduction) across individuals that can disproportionately impact small populations can be grouped into different categories [7]. Demographic stochasticity is caused by random variation in probabilistic birth and death rates resulting in some individuals contributing more offspring or living longer by chance. Extrinsic factors

contribute to environmental stochasticity; spatial or temporal variation in birth and death rates at the population level. In contrast to extrinsic factors and demographic stochasticity, which cause chance variation in realized reproductive rates between individuals, demographic heterogeneity is intrinsic variation in vital rates at the individual level, which may be owing to genetic quality, maternal effects, access to resources or different exposure to stressors [8,9].

Reproductive skew, defined as unequal reproductive success between individuals of the same sex in a population [10,11], is one facet of demographic heterogeneity that may affect extinction risk differently to other variations in reproductive success. Reproductive skew is predicted to affect population growth independently of overall reproductive potential, population age structure and environmental changes [12]. Structured, as opposed to unstructured, variation in vital rates is not independent of the vital rates of other individuals or the identity of the individual [13]. It has been shown that structured demographic heterogeneity that is retained throughout individuals' lifetimes, or 'individual heterogeneity', can have significant effects on extinction risk in small populations [14]. Despite this, population models often incorporate demographic stochasticity, in which all individuals have equal vital rates but the chance of a demographic event is modified by sampling variance, but do not include demographic heterogeneity [15]. Disentangling age-specific reproduction from female reproductive skew and assessing their impacts on population viability will add important dynamics of individual heterogeneity into demographic studies.

For mammals, there is often an assumption that, outside of cooperatively breeding species, there is little reproductive skew among females or it is not important [16,17]. Long-term datasets have shown that it is present in non-cooperatively breeding mammals [18,19] but few studies have investigated the effect on population viability.

Other drivers of population dynamics such as poaching may magnify the effects of female reproductive skew. Poaching has been shown to increase male reproductive skew in African elephants (Loxodonta africana) [20] and catastrophic poaching of male saiga antelopes (Saiga tatarica tatarica) led to a crash in the number of pregnancies owing to disturbed mating behaviour [21]. The combined effect of female reproductive skew and poaching on population viability has yet to be explored. If population persistence depends on the survival and reproduction of a small number of fecund individuals, then the poaching of those individuals may drastically increase extinction risk.

One Critically Endangered subspecies for which assessing extinction risk is vital is the eastern black rhino (Diceros bicornis michaeli). Poaching is still a major threat [22] and Kenya, the major range state for the subspecies, has set a target of 'net growth of at least 5% per annum maintained in at least six established populations' [23, p. 13], which some reserves are not achieving [24]. To minimize poaching risk, Kenyan black rhinos are managed as an artificial meta-population [23]. Isolated populations present the opportunity to conduct proxy natural experiments for studying environmental and demographic drivers of population dynamics. Reproductive skew has been identified within both captive and free-living female black rhinos, including variation in the number of calves, age of first reproduction and inter-calving interval [24–26]. Intrinsic differences in quality may allow particular individuals to benefit from both environmental and social factors. Females are largely solitary but regularly interact with other males and females which have adjacent or overlapping home ranges [27,28]. Dominant females may secure home ranges with better quality diets and breed more successfully. Intraspecies aggression, harassment from males or other females [29,30] and fighting [12], could cause stress and inhibit reproduction [18], with fitter females more likely to successfully harass others while resisting it themselves.

The aim of our study was to assess the local extinction risk of eastern black rhino populations in three Kenyan reserves under multiple drivers of population dynamics including poaching and reproductive skew. We constructed a population viability analysis (PVA), a method which is routinely used to assess the risk of extinction faced by a species or population over a particular time period [31,32] and the importance of particular threats [33]. We defined local extinction as a female population of zero, although populations can become functionally extinct before then. The use of PVAs to effectively quantify extinction risk and predict future population declines is reliant on parameterization from high-quality data and appropriate life-history assumptions [34]. Black rhinos in Kenya are intensively monitored as part of efforts to protect them from poaching and so there are excellent current and historical demographic data available.

We used a data-driven approach to estimate population viability using approximately 40 years of individual-based demographic data. Such long-term datasets are very rare, particularly for free-living populations of a Critically Endangered species. To explore extinction risk, we constructed an age-specific, female-only, individual-based, stochastic population model. An individual-based approach allowed us to include reproductive skew as an intrinsic reproductive score that was relative to the entire population, assigned to each individual at birth and stayed the same through their entire lives. As well as reproductive skew, the model incorporated density dependence, environmental stochasticity and demographic stochasticity. We then simulated population growth over the next 100 years for each reserve. Crucially, we used this as a case study to simulate population growth under different offtake scenarios to assess the effect that reproductive skew and different levels of poaching would have on population viability for a large mammal.

## 2. Methods

### (a) Study populations

We focused on three different Kenyan reserves. The 250 km² Lewa Wildlife Conservancy in Meru County (0.20° N, 37.42° E) founded a 20 km² rhino sanctuary in 1983 and converted completely to a conservancy in 1995. The 360 km² Ol Pejeta Conservancy in Laikipia Country (0.02° N, 36.90° E) founded a rhino sanctuary in 1988. The 235 km² Ol Jogi Conservancy in Laikipia County (0.32° N, 36.98° E) was established as a rhino sanctuary in 1980.

### (b) Data collection

Owing to intensive monitoring to protect from poaching, there are high-quality individual-based demographic data available. Black rhino calves usually stay with their mothers for around 2.5–3 years, making the assignment of maternity almost certain. Paternity is difficult to assign without genetic techniques [35] and we could not include males in this study. However, high skew in

male breeding success [36] suggests that there is a lot of extra breeding potential among males, so generally breeding is unlikely to be limited by male density. In mammalian population modelling, 'female dominance' is often assumed where there are always enough males to fertilize all females [37]. A female-only design was, therefore, appropriate.

The three reserves record the dates of births, deaths, imports (including the ages of imported individuals) and exports. The ages at which females died and gave birth has been accurately recorded since the foundation of each sanctuary. Data are available for the periods 1984–2019 for Lewa, 1980–2019 for Ol Jogi and 1990–2019 for Ol Pejeta.

We constructed a time-to-event demographic dataset for each population documenting whether each female died, bred, or was imported/exported in a given year since they were present in the population. For import and export events, individuals were brought in or removed from the population with no birth or death event. Individuals were only ever translocated once in their lifetimes. Ten females were imported to and 10 exported from Lewa, 19 to and two from Ol Jogi and 18 to and two from Ol Pejeta. Imported and non-imported individuals were not differentiated but we incorporated individual-level differences which capture some of the potential for variation between imported and non-imported individuals in analyses. The demographic outcomes were coded as binary response variables, where 1 indicated birth or survival in a given observation year. Thirty-six calves died before their sex was recorded, eight on Lewa, 25 on Ol Pejeta and three on Ol Jogi. We randomly selected whether each of these was a male or a female and removed the simulated male calves from the model. There were then 99 females recorded from Ol Pejeta, 79 from Lewa, 55 from Ol Jogi and a total of 2252 year-age observations.

## (c) Estimation of age-specific vital rates

We constructed an age-specific model that incorporated vital rate changes across lifespan, an approach recently applied to Asian elephants *(Elephas maximus)* [38]. All analyses used R v. 4.0.1 [39].

Mortality and birth events in the demographic records were used to quantify population vital rates. The proportion of females that died or gave birth to female calves at each specific age provided raw age-specific mortality and birth rates that did not include reproductive skew. The values used in the PVA were estimated from the raw data using generalized additive models (GAMs) implemented in the *mgcv* package [40]. The distributional assumptions were checked using the *DHARMa package* [41] (electronic supplementary material, S1).

## (d) Parameterization of stochastic projection model

To assess the future viability of each population, we built female-only, stochastic individual-based projection models using the predicted age-specific vital rates and projected 100 years into the future. Projections started from the populations present at the end of 2019. We cannot present the age structures of these starting populations owing to confidentiality of black rhino data which was a condition of our research permits and is a policy of the IUCN SSC African Rhino Specialist Group. An individual-based modelling framework allowed us to incorporate demographic stochasticity, an important source of uncertainty, providing an advantage over deterministic models. Every year, individual death and reproduction were simulated using a Bernoulli distribution determined by the probabilities calculated using the GAMs for each reserve (a single trial for each living individual in each year) using the rbinom() function. Reproduction was dependent on survival and occurred after survival/death. We removed individuals over the age of 40 in each year of the simulations (probability of mortality for individuals aged 40 was given a value of 1), because few individuals survived over this age and there was large variation in life-history parameters. No individuals over the age of 40 reproduced in the dataset.

## (e) Estimate of environmental stochasticity

We estimated environmental stochasticity from observed variance in annual vital rates across the study period. We calculated the annual mortality rate for the whole population on each reserve, and the annual birth rate for reproductive ages (5 to 34 years of age) from foundation to 2019. We then calculated the standard deviation of each of these annual vital rates, which represented the environmental stochasticity for each vital rate in each population. To incorporate these into the projections, every year of the simulation we sampled from two truncated normal distributions created using rtruncnorm(). These had a mean of zero, a standard deviation equivalent to that of the annual vital rates and were truncated at 0.5 and −0.5 to prevent unrealistic jumps in population size. The breeding and mortality probability of all individuals were modified separately every year by these simulated factors. After this we ensured that no individuals below the age of 5 years bred, as it was assumed to be a pre-reproductive life stage.

## (f) Estimate of density dependence

While there is evidence for declines in reproduction when black rhinos increase above habitat-specific densities [28], it is uncertain what density-dependent factors regulate their numbers [42,43]. In variable environments like African savannah, carrying capacity is dependent on resource availability [44]. Increasing densities of rhinos can reduce diet availability through browsing pressure [45] but this primarily depends on rainfall [46]. The concept of a fixed ecological carrying capacity (ECC) is, therefore, not particularly meaningful in areas with variable rainfall density [47]. Intra- and intersexual competition may also be important in density-driven changes to vital rates [48]. Male harassment of females can decrease recruitment rates, and variation in fecundity can be driven by sex ratio and density [30]. Social carrying capacity may be a more accurate description of a maximum density of rhinos than ECC. Regardless of the mechanism, black rhino populations cannot grow in a limited area indefinitely, although pre-twentieth century maximum population densities are unknown. Therefore, there must be a way of including density dependence in a biologically relevant way.

Carrying capacity was estimated using the International Union for Conservation of Nature (IUCN) translocation guidelines which incorporate reserve size, average annual rainfall and habitat type [28]. We termed it ECC to be consistent with the IUCN even though it may be determined by both environmental and social factors. All three reserves fall into the category '0.2–0.4 rhino km$^{-2}$'. We assumed that maximum density is 0.4 km$^2$, which gives ECC estimations of 100 for Lewa, 140 for Ol Pejeta and 90 for Ol Jogi. Assuming equal sex ratios, these were halved to give the predicted female ECCs.

We incorporated density dependence into the stochastic projection models using the hypotheses that density dependence only has a significant effect above 0.75(ECC) [28], and populations can increase above the estimated ECC [49]. We also mediated density dependence using environmental stochasticity. Theoretically, large, long-lived species exhibit convex relationships between population size and growth rate [50]. Below 0.75(ECC), environmental stochasticity was calculated as in §2e. If the population was a proportion $x$ above 0.75(ECC), then the distributions from which environmental stochasticity was drawn were altered using $4(x - 0.75)$. The standard deviations were increased by adding $4(x - 0.75)$ to the standard deviation calculated in §2e. The mean of the sampling distributions for breeding and mortality probabilities were decreased or increased from zero by $4(x - 0.75)$, respectively. The distributions were truncated at −0.5 and 0.5 to prevent biologically unrealistic jumps in population size.

## (g) Reproductive skew

We conservatively estimated female reproductive skew by calculating the number of calves each female over the age of 9 years

royalsocietypublishing.org/journal/rspb　Proc. R. Soc. B 289: 20220075

**Table 1.** A summary of the variables included in each analysis.

| | demographic potential growth rate - $r$ | long-term annual population growth rate - $r_{long}$ | effects of reproductive skew and poaching |
|---|---|---|---|
| age-specific vital rates | ✓ | ✓ | ✓ |
| environmental stochasticity | | ✓ | ✓ |
| demographic stochasticity | | ✓ | ✓ |
| density dependence | | ✓ | ✓ |
| reproductive skew | | ✓ | ✓ (with and without) |
| poaching pressure | | | ✓ |

had successfully raised to the age of one year. Generally, the earliest that black rhino females can calve is 5 years old [51], although some females have been recorded to calve between the ages of 4 and 5 years [26]. The number of nulliparous females was dominated by younger individuals, and we considered that including all females between 5 and 9 years would inflate reproductive skew. We chose 9 years as the average age of first calving is around 7 years [51] and 9 years is around the first peak of reproductive probability. The number of yearlings each female had produced was divided by their age above 5 years, to give an annual rate of yearling production.

We calculated a reproductive score using both male and female calves as part of efforts to estimate it conservatively. With a 50 : 50 sex ratio, the skew would be the same as with just female calves. However, owing to the relatively small sample size, using only female calves increased estimates of reproductive skew. This distribution was only used to estimate the reproductive score of females relative to each other, and resulting values were scaled to conserve the average breeding probabilities calculated using the female-only vital rates. Males, therefore, do not feature in the model.

The distribution of reproductive success was created using hist(). At the start of the projection, or at birth, each female was assigned a relative reproductive score, that stayed with them throughout their lives, using the distribution of the rate of yearling production. A value was drawn from this distribution using sample(), assigning each individual an integer of 1 to 10, according to the probabilities from the distribution. All reproductive scores of new individuals were scaled around zero using scale(), to preserve the average breeding probability of each population. Reproductive values were divided by 100 so that the final highest modifications were an order of magnitude lower than the annual breeding probabilities of reproductive age females, which were between 0.1 and 0.2. Every year of the simulation, the probability that each individual female reproduced was altered by their relative reproductive score (electronic supplementary material, S2).

We assumed that the reproductive skew distribution stayed constant. If the female reproductive skew is caused by social interactions such as harassment, or Allee effects [52], then density changes may affect it. However, as we could not attribute definite causes to reproductive variance, we could not confidently alter it as the social context changed.

A formal description of the model, following the protocol set out in Grimm *et al.* [53], can be found in the electronic supplementary material, S2.

## (h) Analysis

### (i) Reproductive skew and poaching
We projected the change in population sizes with and without reproductive skew to test the impact that female reproductive skew has on each population under different poaching regimes.

We used a Bernoulli distribution created with rbinom() to decide whether each adult over the age of 5 years would be poached in a particular year. We refer to this as the percentage annual adult offtake. We increased offtake from 0% to 20%, which allowed us to compare how the probability of extirpation of each population over 100 years changed with and without reproductive skew under different simulated poaching regimes.

### (ii) Long-term growth rates
In order to explore the effect of stochasticity, reproductive skew and density dependence on the projected growth rates, we first estimated the intrinsic rate of increase $r$ of each population using Leslie matrices [37], which we term the demographic potential growth rate. We then explored the difference between demographic potential growth rates and long-run realized population growth rates accounting for environmental stochasticity, demographic stochasticity, density dependence and reproductive skew by calculating the 'long-term annual population growth rate' $r_{long}$ using our stochastic projections. Full methods and mathematical justifications can be found in the electronic supplementary material, S3.

Table 1 shows a summary of the variables included in each analysis. Electronic supplementary material, S4 presents an elasticity analysis.

## 3. Results

## (a) Age-specific demographic parameters and differences between reserves

The observed profile of age-specific birth and mortality rates is typical for long-lived mammals, with relatively high mortality rates for very young and very old individuals (electronic supplementary material, S5). There are four peaks in birth rates, around 8, 15, 23 and 32 years of age, and rapid reproductive senescence after 32 years. The unexpected peak in reproduction just before senescence may be an artefact of lower numbers of older individuals in the dataset. All three reserves exhibited similar age-specific profiles of birth and death rates, but Ol Jogi on average showed higher average mortality rates and lower birth rates than the other two, where birth rates easily exceeded mortality rates (electronic supplementary material, S5).

A comparison of $r$ and $r_{long}$ values highlights differences between the reserves and the effect of stochasticity and reproductive skew on the projected growth rates (table 2). Including these processes decreases the growth rates from the demographic potential of all three populations, and for Ol Jogi a positive $r$ was accompanied by a negative $r_{long}$ over 100 years. Lewa displayed the highest demographic potential

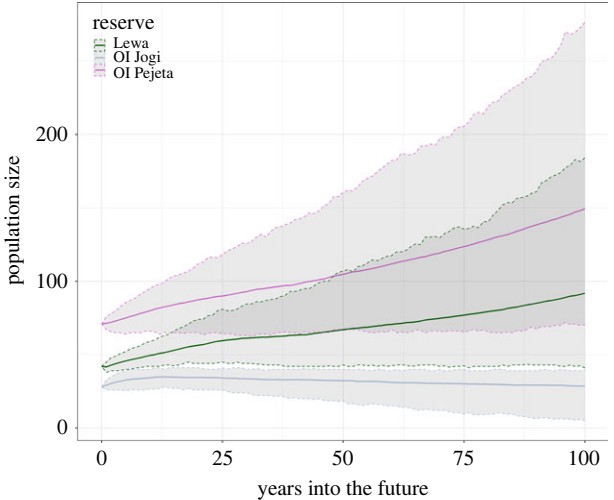

**Figure 1.** Baseline projections with reproductive skew. Projected changes in the sizes of the populations 100 years into the future. The solid lines show the mean of all 500 simulations and dotted lines show 95% confidence intervals.

**Table 2.** Intrinsic rates of increase ($r$) (calculated from Leslie matrices) and estimated population growth rates calculated from the simulations ($r_{long}$) with 95% confidence intervals.

| reserve | $r$ | $r_{long}$ |
|---------|-----|------------|
| Lewa | 0.0555 | $0.0055 \pm 0.0034$ |
| Ol Jogi | 0.0196 | $-0.0014 \pm 0.0058$ |
| Ol Pejeta | 0.0547 | $0.0066 \pm 0.0028$ |

growth rate, Ol Pejeta the highest long-term growth rate, and Ol Jogi had much lower values for both.

The ranges and means of population sizes of projections with stochasticity and reproductive skew across 500 simulations varied across reserves (figure 1). On average, the populations on Lewa and Ol Pejeta are predicted to continue to increase over 100 years, exceeding the predicted ECCs of 50 and 70, respectively. On Ol Jogi, the population is predicted to decline slowly.

## (b) Reproductive skew and population dynamics

There was substantial skew in reproductive success across individuals and there were different patterns of skew between the reserves (electronic supplementary material, S6). The overall reproductive skew distribution shows that black rhino females in these populations do vary in their reproductive success and suggests not all the female reproductive potential is being realized. This was most strongly evident on Ol Pejeta and then Ol Jogi, whereas Lewa's distribution was more symmetric around the mean.

Generally, including reproductive skew in the model has a significant effect on the projected change in the population (figure 2). With reproductive skew included the population size on Ol Jogi is lower than the 2019 size on average after 100 years. There were lesser effects on Lewa and Ol Pejeta (electronic supplementary material, S7). Female reproductive skew can increase the extinction probability for small populations, or those with low intrinsic growth rates, even without offtake.

Offtake had expected negative impacts on population persistence, with a greater than 50% probability of extinction

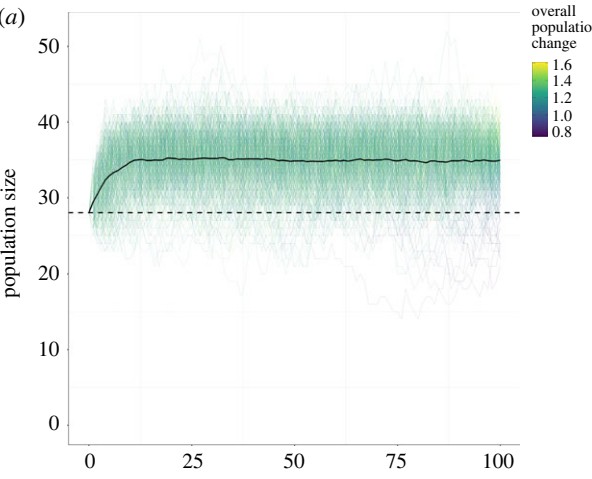

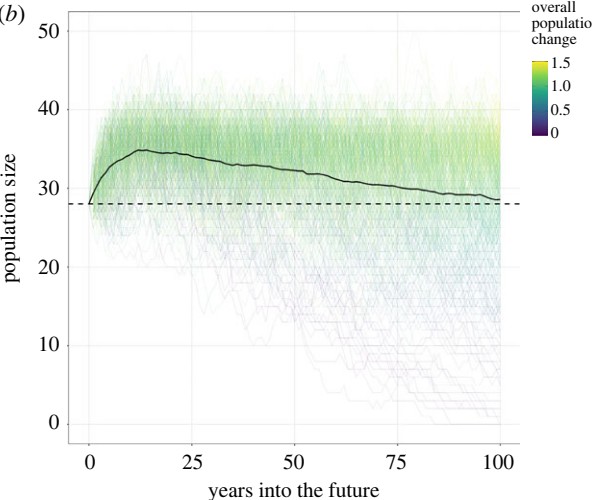

**Figure 2.** A comparison of the population projections for Ol Jogi (*a*) without and (*b*) with female reproductive skew. Graphs for Lewa and Ol Pejeta can be found in the electronic supplementary material, S7. (Online version in colour.)

when adult offtake was greater than 4% (Ol Jogi), 11% (Lewa), or 12% (Ol Pejeta). Crucially, the inclusion of reproductive skew significantly increased the probability of extinction (figure 3). Even with no offtake, 1.2% of the Ol Jogi simulations went extinct over 100 years when reproductive skew was included. At 5% offtake, the probability increases from 13.6% to 77.8%. The larger populations were also affected by reproductive skew. At 10% offtake, the extinction probabilities on Ol Pejeta and Lewa increased from 0.4% and 2.2% to 10.4% and 44.4% respectively when reproductive skew was included in the models.

## 4. Discussion

Reproductive skew is an important factor to consider in studies of population dynamics [13] but it is rarely incorporated into PVAs. We provide evidence that variation in female breeding success can increase extinction risk from poaching. Datasets that allow for the estimation of variation in breeding success are rare, but PVAs which do not include it may be underestimating extinction risk. This work also highlights important differences between three key Kenyan black rhino reserves.

The estimated maximum long-term intrinsic growth rate of a black rhino population is 9–11% pa [43]. To achieve

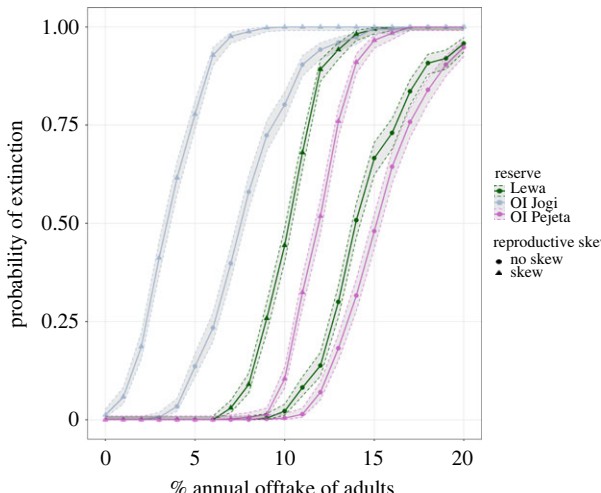

**Figure 3.** The change in proportion of 500 simulations that reach a population size of 0 over 100 years with and without reproductive skew under different levels of offtake. The 95% confidence intervals were calculated using the adjusted Wald technique. (Online version in colour.)

this, and allow the use of 'established populations as a 'breeding bank' … to build up other populations and to expand into new secure areas with suitable habitats' [23, p. 13], the underlying causes of variation must be identified.

The vulnerability of the Ol Jogi population to extinction when faced with stochasticity and female reproductive skew is likely to be a consequence of both its smaller population size [54] and lower intrinsic growth rate. The greater variability in the long-term annual population growth rate ($r_{long}$) of Ol Jogi is probably owing to the long period that the projections spend oscillating close to a maximum size. We ran projections using Ol Jogi vital rates but the starting population structure and carrying capacity of Ol Pejeta and no simulations went extinct but the mean population size was significantly lower than the other two reserves, at around 54 after 100 years. As this study focused on real-world populations, we did not vary population size, growth rates, or reproductive skew to investigate how their impact on extinction risk varied with their magnitude. Future work with simulated populations could vary these factors independently to test their relative importance.

Determining the cause of reserve-level differences is crucial for the conservation of black rhinos in Kenya. One possibility is that population performance is affected by differences in historical reproductive skew. However, Ol Pejeta seems to have the greatest skew, and the highest proportion of nulliparous or very slow breeding females (electronic supplementary material, S6). Differences in habitat variability could cause population-level differences. However, black rhino populations are relatively resilient to drought, potentially owing to their browsing diet [12,55]. Our estimations of environmental stochasticity are based on observed variations in annual vital rates, but it could be that our model over emphasizes the role of environmental variation when social context could be the cause. Assigning causality to temporal variations in vital rates is a key challenge for future research.

While the study reserves contain broadly similar habitat, demographic differences could be caused by fine-scale habitat differences. This includes disease [56,57], predation [58] and diet availability. Assessments estimate that Ol Jogi has a higher proportion of browse that is considered highly suitable

for black rhinos, so its lower intrinsic growth rate is surprising [59]. While there is evidence for black rhino dietary preferences [60,61], new methods including metabarcoding [62] could be used to directly link dietary composition with fitness. Although *D. b. michaeli* has retained the most genetic diversity of the remaining subspecies of black rhino [63], lower heterozygosity has been linked to reduced male reproductive success [35]. Low diversity could be affecting populations in unknown ways, and there are suspicions that bilateral blindness in Kenyan black rhinos could have a genetic component.

Large mammal species can be considered to be meta-populations if they have discreet breeding subpopulations with different growth rates and demographic fates [64]. We provide evidence that these black rhino populations fulfil these criteria. The physical, social and economic infrastructure required to maintain rhino reserves makes it unfeasible to move all individuals to optimal habitat. Also, maintaining populations in different habitat types reduces overall extinction risk of a metapopulation [65]. Biological management, including the translocation of high-value females that takes accounts for genetic factors, could lower the risk of extinction on reserves like Ol Jogi. If future work can explain the differences in population growth rates between the three reserves, then conservation planning for black rhinos could take account of factors that increase death rates and decrease birth rates.

Male reproductive skew has been found to have a small impact on the extinction risk of mammal populations affected by poaching [66]. Although male reproductive skew does not directly impact population dynamics, it may have impacts on long-term genetic variation [67]. It should be noted that lower effective population size caused by reproductive skew in black rhinos may be compensated for to some extent by higher heterozygosity of dominant males [35]. We assumed that male availability does not affect breeding rates, but it could be important at low densities. It is likely that low densities of males and difficulties finding mates would exacerbate female reproductive skew and its population level effects, which would not invalidate our conclusions.

Female reproductive skew may be important in the short and long term even though it is often overlooked in non-cooperatively breeding species. We show that reproductive skew can affect projected population growth, particularly in small populations or those with low intrinsic growth rates, and provide empirical evidence for the theoretical proposition that structured variation in fecundity probabilities can increase extinction risk [68]. Even if PVAs are overly pessimistic [69], understanding the additive impact of skew on stochasticity is fundamentally important. It has far-reaching implications for conservation and the estimation of extinction risk, particularly for species that are affected by offtake.

As far as we are aware, the combined impact of variation in female breeding success and poaching has not been investigated in a large-bodied vertebrate species. It has been found that poaching female adult giant pandas (*Ailuropoda melanoleuca*), rather than adult males or young individuals, leads to lower population sizes and a higher chance of extinction [70]. Here we have found that the level of female reproductive skew present in our study populations significantly increases the extinction risk of populations of large mammals that are affected by poaching. This may be because a large proportion of total reproductive potential is invested in relatively few individuals. Without poaching, these very fit individuals may prevent extinction [14] and we would expect 'frail'

royalsocietypublishing.org/journal/rspb　Proc. R. Soc. B 289: 20220075

individuals to be disproportionately lost, leading to higher growth rates and more resilient cohorts over time [8,9]. Poaching, however, is not selective and is likely to remove fit individuals. This will have a big impact on the growth of the population and destabilize cohorts. A crucial research priority is to find the level of female reproductive skew which starts to impact population dynamics. This will vary depending on environmental and social factors, but an estimated threshold would be very useful for management. Datasets that allow for the estimation of variation in breeding success are rare. However, we suggest that the lack of research into the combination of poaching and reproductive skew on population dynamics may cause the underestimation of extinction risk for populations affected by poaching.

With respect to the conservation of black rhinos, the importance of female reproductive skew has two implications. Firstly, although many factors influence extinction risk, even low levels of poaching have the potential to damage the long-term viability of black rhino populations [71]. Tackling poaching is already a priority for Kenya, which aims to keep levels less than 1% per annum [23]. At 1% offtake in our projections with reproductive skew, no simulations reached extinction over 100 years on Lewa or Ol Pejeta, but almost 5% did so on Ol Jogi. Secondly, management should be used to reduce female reproductive skew and encourage as many females as possible to breed. The methods will depend on the causes of reproductive skew, whether fitter females are able to monopolize the best resources, resist disease and predation, take dominant roles in social hierarchies or resist harassment from others. Future research should focus on factors that decrease the probability of females breeding and raising young, especially drivers of nulliparity or persistently poor breeding performance. For example, if reproductive skew is largely caused by differences in diet quality on the home-range scale, then habitat management could be used. This may be difficult in many places, especially if the abundance of preferred food plants is limited by rainfall [46], and supplementary feeding is often controversial. Placing new reserves in optimal habitat may be more feasible. On the other hand, if male harassment is preventing breeding in male-biased populations, then strategic metapopulation management including the removal of males could even out the sex ratio. While translocations pose risks, biological management of black rhinos is routine. The difficulty lies in where to put excess males if many reserves struggle with male-biased sex ratios.

Understanding the causes of reproductive skew will allow for it to change as a function of population features in population models. Poaching may actually set up feedback loops that worsen reproductive skew and lead to faster population declines. Poaching had indirect effects on demography in Kruger National Park owing to reduced mate-finding as an Allee effect, disturbed social dynamics or increased calf predation [52]. Although our study reserves are much smaller than Kruger, female black rhinos change their spatial organization very slowly after the death of a neighbouring individual [30], so poaching may decrease encounters with males and extended re-establishment of male dominance may make females reluctant to mate [52].

The demographic importance of female reproductive skew poses a difficult problem for the conservation of other species. Assessing whether a species exhibits reproductive skew requires long-term data collection on an individual level, which is difficult and expensive. Designing conservation programmes to mitigate the impact of female reproductive skew is even more challenging. Apart from tackling poaching and providing optimal habitat conditions, conservation specifically focused on alleviating reproductive skew must be done on an individual basis, including encouraging reproduction in females with low success or strategic biological management of metapopulations. The individual-level data and monitoring available for black rhino provide a way forward for assessing and mitigating the effect of skew but represent important knowledge gaps in the conservation of other species.

**Ethics.** This project was approved by the University of Manchester's Committee for the ethical review of category D research (ref: 0030). Research was conducted in affiliation with the Kenya Wildlife Service and licensed by the Republic of Kenya's National Commission for Science and Innovation (permit nos.: NACOSTI/P/17/87006/16178 and NACOSTI/P/19/1947).

**Data accessibility.** Black rhino data are treated as sensitive and confidential and we are not able to provide the raw data that this work is based on. In lieu of this, we have provided dummy data in the supplementary information, in the same format as the real data, that can be used the emulate the analyses. Access to the data neccessary to replicate this study must be agreed by the individual study reserves. Readers must send data requests to the corresponding author and these will be passed on to reserve management teams. Please note that access to the data will require research permits and a data use agreement.

The dummy data are provided in the electronic supplementary material [72].

**Authors' contributions.** N.H.S.: conceptualization, data curation, formal analysis, investigation, methodology, project administration, validation, visualization, writing—original draft, writing—review and editing; J.J.: formal analysis, methodology, resources, validation, writing—review and editing; G.C.: data curation, writing—review and editing; J.G.: data curation, writing—review and editing; D.K.: data curation, writing—review and editing; S.M.: data curation, writing—review and editing; S.N.: data curation, writing—review and editing; S.S.: conceptualization, methodology, supervision, writing—review and editing.

All authors gave final approval for publication and agreed to be held accountable for the work performed therein.

**Conflict of interest declaration.** We declare we have no competing interests.

**Funding.** N.H.S. is funded by the NERC Manchester-Liverpool DTP, NE/L002469/1, and Chester Zoo as part of their Conservation Scholars programme.

**Acknowledgements.** We express sincere thanks to the security, monitoring and research teams on all three reserves who meticulously collected the demographic data, especially I. Lemaiyan who was integral to this on Lewa. We would also like to thank F. Omengo and KWS for providing permission to conduct this study and to M. Kamau, A. Kibungei and B. Gituku for comments on an early version of the manuscript.

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
