## [Peer Review File · Proceedings of the Royal Society B: Biological Sciences]

Review History

RSPB-2021-1405.R0 (Original submission)

Review form: Reviewer 1

Recommendation

Accept with minor revision (please list in comments)

Scientific importance: Is the manuscript an original and important contribution to its field?

Excellent

General interest: Is the paper of sufficient general interest?

Excellent

Quality of the paper: Is the overall quality of the paper suitable?

Excellent

Is the length of the paper justified?

Yes

Should the paper be seen by a specialist statistical reviewer?

No

Do you have any concerns about statistical analyses in this paper? If so, please specify them explicitly in your report.

No

It is a condition of publication that authors make their supporting data, code and materials available - either as supplementary material or hosted in an external repository. Please rate, if applicable, the supporting data on the following criteria.

Is it accessible?

N/A

Is it clear?

N/A

Is it adequate?

N/A

Do you have any ethical concerns with this paper?

No

Comments to the Author

I have attached a file listing my concerns and suggestions for the authors. (See Appendix A)

Review form: Reviewer 2 (Sam Ferreira)

Recommendation

Major revision is needed (please make suggestions in comments)

Scientific importance: Is the manuscript an original and important contribution to its field?

Acceptable

General interest: Is the paper of sufficient general interest?

Marginal

Quality of the paper: Is the overall quality of the paper suitable?

Acceptable

Is the length of the paper justified?

Yes

Should the paper be seen by a specialist statistical reviewer?

Yes

Do you have any concerns about statistical analyses in this paper? If so, please specify them explicitly in your report.

No

It is a condition of publication that authors make their supporting data, code and materials available - either as supplementary material or hosted in an external repository. Please rate, if applicable, the supporting data on the following criteria.

Is it accessible?

N/A

Is it clear?

N/A

Is it adequate?

N/A

Do you have any ethical concerns with this paper?

No

Comments to the Author

Paper Review: Female reproductive skew exacerbates the extinction risk from poaching in eastern black rhino

The authors provide an assessment of the consequences of reproductive skew in females for rhino persistence in the presence of various levels of poaching pressure. I provide conceptual comments aimed at adding value to the manuscript.

Comments:

- 1) Considering reproductive skew is important for eastern black rhino given that most populations are small and would be vulnerable to stochastic processes. The manuscript could benefit from perhaps highlighting stochastic risk associated with the persistence of small populations. Reproductive skew is one of those stochastic risks that realize whether a species is social or not. This could compliment the existing rationale.
- 2) One aspect that would improve the manuscript is a more explicit definition of reproductive skew. The present definition provided in lines 72-73 is confusing – that definitions comprise various aspects of vital rates (birth rate, survival during the suckling period etc.). This contrast an understanding that reproductive skew is a partitioning of reproduction among the same sex.
- 3) A third element of clarification is to provide some prediction of what measures can reflect reproductive skew that makes it different from simple variability in reproductive output as indicated that associate with age, environmental drivers (e.g. see papers on drought effects on birth rates), synchronization of reproduction and variability within an individual cow.
- 4) The authors could add additional value in the discussion on the possible consequences of reproductive skew for meta-population dynamics, particularly if future rhino management seek actively to mimic meta-population processes. For instance, Eastern black rhino occur in relatively small-protected areas that reflects patches of suitable habitat embedded in a landscape of unsuitable habitat. A species may persist locally and evolutionary if they overcome the risks imposed by landscape fragmentation of suitable habitat. In such instances populations, albeit small, that occasionally interact with other populations create a meta-population that has, as a whole, a higher chance of persistence than individual populations on their own. Meta-populations typically (1) span landscapes that have patches of suitable habitat supporting local breeding populations, with vacant habitat patches that individuals can colonize; (2) suitable habitat vary on those patches resulting in different birth and death rates in asynchrony between patches; (3) colonization and extinction of species on a patch takes place; and (4) dispersal occurs between local populations occurring on patches. For large mammals, the time- and spatial-scale over which population dynamics play out results in more lenient meta-population criteria including (1) local breeding populations should be discrete rather than inhabiting discrete habitat patches; and (2) local populations should have dissimilar growth rates. The Eastern black rhinos may thus function as a meta-population within Kenya. The present analyses treated each population in isolation and defined how reproductive skew under various levels of poaching impact extinction risk of each population. Reflecting on how reproductive skew under different levels of poaching impact extinction risk of the meta-population could add additional value to the manuscript and practical implications for conservation managers.
- 5) Some elements that are missing are the mechanisms that could lead and maintain reproductive skew. Allee effects of reduced mating opportunities for instance etc. This is important because the key premise of the future predictions is that the patterns of reproductive skew remains constant over a 100 year period in both the individual and matrix population models.

- 6) It could be useful to have a figure or table that illustrate the observed reproductive skew in the three populations. At present, this resides in the supplementary material.
- 7) Some clarity is needed with regards to long term growth estimates derived from year 25 to year 100. The analyses modelled density-dependence. This means that population growth will not be constant over time. It is thus hard to interpret the meaning of the growth rate comparisons.
- 8) Some additional considerations on demographic responses as population increase could be useful that considers sequential responses of calf survival, fecundity and then adult survival (see Eberhardt, L.L., 2002. A paradigm for population analysis of long-lived vertebrates. *Ecology*, 83(10), pp.2841-2854, as well as Sibly, R.M., Barker, D., Denham, M.C., Hone, J. and Pagel, M., 2005. On the regulation of populations of mammals, birds, fish, and insects. *Science*, 309(5734), pp.607-610.). This could provide an alternative approach to estimate ECC based on the responses noted by rhinos in the data.
- 9) The first part of the discussion may be best integrated into a short description of the study sites.
- 10) It could be useful to also add in the discussion some reflection of sensitivity of population persistence of reproductive skew in large populations under different poaching levels in combination with an element of meta-population consequences.
- 11) The discussion do provide two "management recommendations", but could benefit from including more detail on mechanisms that can lead to reproductive skew and then how to explicitly address a mechanism (see Ferreira, S.M., le Roex, N. and Greaver, C., 2019. Species-specific drought impacts on black and white rhinoceroses. *Plos one*, 14(1), p.e0209678; le Roex, N. and Ferreira, S.M., 2021. Rhino birth recovery and resilience to drought impact. *African Journal of Ecology*, <https://doi.org/10.1111/aje.12854>; le Roex, N. and Ferreira, S.M., 2020. Age structure changes indicate direct and indirect population impacts in illegally harvested black rhino. *PLoS one*, 15(7), p.e0236790).
- 12) Overall, this paper has an important contribution to be made, but requires differentiation between consequences of small populations, demographic variation and reproductive skew as a component of demographic variation.

Decision letter (RSPB-2021-1405.R0)

19-Jul-2021

Dear Mr Harvey:

I am writing to inform you that your manuscript RSPB-2021-1405 entitled "Female reproductive skew exacerbates the extinction risk from poaching in the eastern black rhino" has, in its current form, been rejected for publication in *Proceedings B*.

This action has been taken on the advice of referees, who have recommended that substantial revisions are necessary. With this in mind we would be happy to consider a resubmission, provided the comments of the referees are fully addressed. However please note that this is not a provisional acceptance.

Sincerely,
Dr Daniel Costa
mailto:proceedingsb@royalsociety.org

Associate Editor

Comments to Author:

Two referees reviewed your manuscript and feel that it could constitute a valuable contribution but they request clarification on a range of very important issues. In particular, your interpretation of what 'reproductive skew' is lacks clarity. Referee 1 makes a very important distinction between individual heterogeneity in vital rates across individuals (demographic stochasticity) and heterogeneity in vital rates among age/stage classes, which can be incorporated in purely deterministic models. Referee 2 seems to assume that your analysis considers reproductive skew as one of the many stochastic risks faced by small populations. Your analyses first estimate age-specific vital rates and then use them to implement the projection model using an individual-based approach. Thus, in your case, it would seem as if reproductive skew is directly determined by the structured fecundity of your model and does not amount to a stochastic risk. This is an important observation as one could think of reproductive skew as a purely stochastic process whereby a fraction of individuals in the population (which each individual chosen randomly) can contribute the most to reproduction regardless of their age/stage.

Another very important source of confusion is the lack of a clear presentation of the assumptions of your projection model. In particular, referee 1 rightly points out in the main text, you indicate that the order of events is survival-> reproduction but in S2 you state the opposite is true. Overall, your presentation of the projection model is rather poor. It would help to include a figure with a flow chart showing the sequence of events that individuals undergo and the underlying distribution from which probabilities of survival and reproduction are drawn.

Both referees provide many other important comments that need to be carefully taken into consideration by either incorporating them into the manuscript or providing a very clear explanation of why you didn't do so.

Reviewer(s)' Comments to Author:

Referee: 1

Comments to the Author(s)

I have attached a file listing my concerns and suggestions for the authors.

Referee: 2

Comments to the Author(s)

Paper Review: Female reproductive skew exacerbates the extinction risk from poaching in eastern black rhino

The authors provide an assessment of the consequences of reproductive skew in females for rhino persistence in the presence of various levels of poaching pressure. I provide conceptual comments aimed at adding value to the manuscript.

Comments:

- 1) Considering reproductive skew is important for eastern black rhino given that most populations are small and would be vulnerable to stochastic processes. The manuscript could benefit from perhaps highlighting stochastic risk associated with the persistence of small populations. Reproductive skew is one of those stochastic risks that realize whether a species is social or not. This could compliment the existing rationale.
- 2) One aspect that would improve the manuscript is a more explicit definition of reproductive skew. The present definition provided in lines 72-73 is confusing – that definitions comprise various aspects of vital rates (birth rate, survival during the suckling period etc.). This contrast an understanding that reproductive skew is a partitioning of reproduction among the same sex.
- 3) A third element of clarification is to provide some prediction of what measures can reflect reproductive skew that makes it different from simple variability in reproductive output as indicated that associate with age, environmental drivers (e.g. see papers on drought effects on birth rates), synchronization of reproduction and variability within an individual cow.
- 4) The authors could add additional value in the discussion on the possible consequences of reproductive skew for meta-population dynamics, particularly if future rhino management seek actively to mimic meta-population processes. For instance, Eastern black rhino occur in relatively small-protected areas that reflects patches of suitable habitat embedded in a landscape of unsuitable habitat. A species may persist locally and evolutionary if they overcome the risks imposed by landscape fragmentation of suitable habitat. In such instances populations, albeit small, that occasionally interact with other populations create a meta-population that has, as a whole, a higher chance of persistence than individual populations on their own. Meta-populations typically (1) span landscapes that have patches of suitable habitat supporting local breeding populations, with vacant habitat patches that individuals can colonize; (2) suitable habitat vary on those patches resulting in different birth and death rates in asynchrony between patches; (3) colonization and extinction of species on a patch takes place; and (4) dispersal occurs between local populations occurring on patches. For large mammals, the time- and spatial-scale over which population dynamics play out results in more lenient meta-population criteria including (1) local breeding populations should be discrete rather than inhabiting discrete habitat patches; and (2) local populations should have dissimilar growth rates. The Eastern black rhinos may thus function as a meta-population within Kenya. The present analyses treated each population in isolation and defined how reproductive skew under various levels of poaching impact extinction risk of each population. Reflecting on how reproductive skew under different levels of poaching impact extinction risk of the meta-population could add additional value to the manuscript and practical implications for conservation managers.
- 5) Some elements that are missing are the mechanisms that could lead and maintain reproductive skew. Allee effects of reduced mating opportunities for instance etc. This is important because the key premise of the future predictions is that the patterns of reproductive skew remains constant over a 100 year period in both the individual and matrix population models.
- 6) It could be useful to have a figure or table that illustrate the observed reproductive skew in the three populations. At present, this resides in the supplementary material.
- 7) Some clarity is needed with regards to long term growth estimates derived from year 25 to year 100. The analyses modelled density-dependence. This means that population growth will not be constant over time. It is thus hard to interpret the meaning of the growth rate comparisons.
- 8) Some additional considerations on demographic responses as population increase could be useful that considers sequential responses of calf survival, fecundity and then adult survival (see Eberhardt, L.L., 2002. A paradigm for population analysis of long-lived vertebrates. *Ecology*,

83(10), pp.2841-2854, as well as Sibly, R.M., Barker, D., Denham, M.C., Hone, J. and Pagel, M., 2005. On the regulation of populations of mammals, birds, fish, and insects. *Science*, 309(5734), pp.607-610.). This could provide an alternative approach to estimate ECC based on the responses noted by rhinos in the data.

9) The first part of the discussion may be best integrated into a short description of the study sites.

10) It could be useful to also add in the discussion some reflection of sensitivity of population persistence of reproductive skew in large populations under different poaching levels in combination with an element of meta-population consequences.

11) The discussion do provide two “management recommendations”, but could benefit from including more detail on mechanisms that can lead to reproductive skew and then how to explicitly address a mechanism (see Ferreira, S.M., le Roex, N. and Greaver, C., 2019. Species-specific drought impacts on black and white rhinoceroses. *Plos one*, 14(1), p.e0209678; le Roex, N. and Ferreira, S.M., 2021. Rhino birth recovery and resilience to drought impact. *African Journal of Ecology*, <https://doi.org/10.1111/aje.12854>; le Roex, N. and Ferreira, S.M., 2020. Age structure changes indicate direct and indirect population impacts in illegally harvested black rhino. *PloS one*, 15(7), p.e0236790).

12) Overall, this paper has an important contribution to be made, but requires differentiation between consequences of small populations, demographic variation and reproductive skew as a component of demographic variation.

Author's Response to Decision Letter for (RSPB-2021-1405.R0)

See Appendix B.

RSPB-2022-0075.R0

Review form: Reviewer 2

Recommendation

Accept with minor revision (please list in comments)

Scientific importance: Is the manuscript an original and important contribution to its field?

Good

General interest: Is the paper of sufficient general interest?

Acceptable

Quality of the paper: Is the overall quality of the paper suitable?

Good

Is the length of the paper justified?

Yes

Should the paper be seen by a specialist statistical reviewer?

No

Do you have any concerns about statistical analyses in this paper? If so, please specify them explicitly in your report.

No

It is a condition of publication that authors make their supporting data, code and materials available - either as supplementary material or hosted in an external repository. Please rate, if applicable, the supporting data on the following criteria.

Is it accessible?

Yes

Is it clear?

Yes

Is it adequate?

Yes

Do you have any ethical concerns with this paper?

No

Comments to the Author

Dear Editor,

I previously reviewed this paper. The authors have addressed several concerns well in the revision.

Although the focus is on black rhino, the outcomes of reproductive skew on dynamics of large mammal species carry broader importance. The benefits of large mammal species to ecosystems and people further get degraded by various global environmental change drivers. Understanding the consequences of reproductive skew provide opportunities for authorities to anticipate and respond.

Some small comments:

Lines 106 to 114 - Perhaps reflect extinction risk at a local scale - extinction risk of a local population of black rhino. Extinction risk should be defined here already - I imagine the authors focus on functional extinction. Perhaps clarify how PVA considers this local functional extinction risk.

Line 123 to 125 - This is an important focus, but could also be done without using any empirical data. Perhaps the authors can clarify the use of a specific case study as they have done.

Lines 140-142 - The authors make clear their assumptions. Male availability, however, could be key at low densities. Perhaps reflect the potential risk of this assumption in the discussion and dismiss it as a concern for the three case studies.

Lines 164 to 167 - Perhaps it will add value if the authors include the definition of age-specific mortality and birth rates. For example calves (was it males and females) born per female of age x? It is not clear if this is now a summary variable disregarding reproductive skews?

Lines 187 to 195 - The authors should reflect on papers on drought responses and resilience of black rhinos to environmental perturbations - limited responses and resistance to drought suggest the authors over-emphasize the role of environmental stochasticity - perhaps reflect in the discussion. See papers by le Roex, N. and Ferreira, S.M., 2021. Rhino birth recovery and resilience to drought impact. *African Journal of Ecology*, 59(2), pp.544-547. and Ferreira, S.M., le Roex, N. and Greaver, C., 2019. Species-specific drought impacts on black and white rhinoceroses. *Plos one*, 14(1), p.e0209678.

Lines 214 to 225 - The authors should reflect on Social Carrying Capacity given the reflections on intra-specific harassment and then rationalize that the ECC approach may also reflect a social carrying capacity concept.

Lines 271 to 280 - the description highlight four variables' influence benchmarked against a best model. In effect is there a fifth variable as well which is poaching pressure. This is a complex set of combinations and perhaps the authors could clarify in a summary table.

Table 1 - are there any confidence intervals?

Figure 3 - Perhaps the authors should reflect what the level of reproductive skew was and whether this did associate with the magnitude of shift in probability of extinction as % annual off take of adults increased. One may ask, at what threshold of reproductive skew are effects negligible irrespective of demographic, environmental and density-dependence factors. But I do note some observations in Lines 354 to 357. This will have impact on the discussion in lines 391 to 406. It does appear that several other factors may influence the sensitivity of population performance to the interaction between reproductive skew and poaching pressure. The authors will need to reflect on this.

Lines 348-349 - this could be effectively two additional factors that influence growth on top of the four or five described in the methods. Perhaps clarify why not formally included.

Lines 408 to 410 - Note comment above - numerous other influences can complicate this result and statement. Perhaps tone down.

Lines 411 to to 421 introduce various biological management options which would vary in feasibility from place to place. Perhaps a sentence or two on feasibility and logistical considerations.

Lines 424 to 427 - correct. Poaching was speculated to influence indirectly on dynamics of black rhino in Kruger.

Decision letter (RSPB-2022-0075.R0)

11-Feb-2022

Dear Dr Harvey Sky:

Your manuscript has now been peer reviewed and the reviews have been assessed by an Associate Editor. The reviewers' comments (not including confidential comments to the Editor) and the comments from the Associate Editor are included at the end of this email for your reference. As you will see, the reviewers and the Editors have raised some concerns with your manuscript and we would like to invite you to revise your manuscript to address them.

To submit your revision please log into <http://mc.manuscriptcentral.com/prsb> and enter your Author Centre, where you will find your manuscript title listed under "Manuscripts with

Decisions." Under "Actions", click on "Create a Revision". Your manuscript number has been appended to denote a revision.

Research ethics:

Use of animals and field studies:

It is a condition of publication that you make available the data and research materials supporting the results in the article (<https://royalsociety.org/journals/authors/author-guidelines/#data>). Datasets should be deposited in an appropriate publicly available repository and details of the associated accession number, link or DOI to the datasets must be included in the Data Accessibility section of the article (<https://royalsociety.org/journals/ethics-policies/data-sharing-mining/>). Reference(s) to datasets should also be included in the reference list of the article with DOIs (where available).

All supplementary materials accompanying an accepted article will be treated as in their final form. They will be published alongside the paper on the journal website and posted on the online figshare repository. Files on figshare will be made available approximately one week before the

accompanying article so that the supplementary material can be attributed a unique DOI. Please try to submit all supplementary material as a single file.

Please submit a copy of your revised paper within three weeks. If we do not hear from you within this time your manuscript will be rejected. If you are unable to meet this deadline please let us know as soon as possible, as we may be able to grant a short extension.

Best wishes,
Dr Locke Rowe
mailto:proceedingsb@royalsociety.org

Associate Editor

Comments to Author:

Your resubmission was evaluated by one of the referees who reviewed your original submission. Overall, the resubmitted manuscript represents a great improvement over the original version. There are, nevertheless, some need for further clarifications as per the referee comments. From my part I would like to highlight that you seem to use “reproductive skew” and “structured demographic heterogeneity” as if they were exchangeable terms but they are not. Reproductive skew is one type of “structured demographic heterogeneity”. To avoid confusion, please remove “defined as structured demographic heterogeneity” in line 76 and replace “structured demographic heterogeneity” in line 119 with “reproductive skew”. This also applies to line 395.

Reviewer(s)' Comments to Author:

Referee: 2

Comments to the Author(s).

Dear Editor,

I previously reviewed this paper. The authors have addressed several concerns well in the revision.

Although the focus is on black rhino, the outcomes of reproductive skew on dynamics of large mammal species carry broader importance. The benefits of large mammal species to ecosystems and people further get degraded by various global environmental change drivers.

Understanding the consequences of reproductive skew provide opportunities for authorities to anticipate and respond.

Some small comments:

Lines 106 to 114 - Perhaps reflect extinction risk at a local scale - extinction risk of a local population of black rhino. Extinction risk should be defined here already - I imagine the authors focus on functional extinction. Perhaps clarify how PVA considers this local functional extinction risk.

Line 123 to 125 - This is an important focus, but could also be done without using any empirical data. Perhaps the authors can clarify the use of a specific case study as they have done.

Lines 140-142 - The authors make clear their assumptions. Male availability, however, could be key at low densities. Perhaps reflect the potential risk of this assumption in the discussion and dismiss it as a concern for the three case studies.

Lines 164 to 167 - Perhaps it will add value if the authors include the definition of age-specific mortality and birth rates. For example calves (was it males and females) born per female of age x? It is not clear if this is now a summary variable disregarding reproductive skews?

Lines 187 to 195 - The authors should reflect on papers on drought responses and resilience of black rhinos to environmental perturbations - limited responses and resistance to drought suggest the authors over-emphasize the role of environmental stochasticity - perhaps reflect in the discussion. See papers by le Roex, N. and Ferreira, S.M., 2021. Rhino birth recovery and resilience to drought impact. *African Journal of Ecology*, 59(2), pp.544-547. and Ferreira, S.M., le Roex, N. and Greaver, C., 2019. Species-specific drought impacts on black and white rhinoceroses. *Plos one*, 14(1), p.e0209678.

Lines 214 to 225 - The authors should reflect on Social Carrying Capacity given the reflections on intra-specific harassment and then rationalize that the ECC approach may also reflect a social carrying capacity concept.

Lines 271 to 280 - the description highlight four variables' influence benchmarked against a best model. In effect is there a fifth variable as well which is poaching pressure. This is a complex set of combinations and perhaps the authors could clarify in a summary table.

Table 1 - are there any confidence intervals?

Figure 3 - Perhaps the authors should reflect what the level of reproductive skew was and whether this did associate with the magnitude of shift in probability of extinction as % annual off take of adults increased. One may ask, at what threshold of reproductive skew are effects negligible irrespective of demographic, environmental and density-dependence factors. But I do note some observations in Lines 354 to 357. This will have impact on the discussion in lines 391 to 406. It does appear that several other factors may influence the sensitivity of population performance to the interaction between reproductive skew and poaching pressure. The authors will need to reflect on this.

Lines 348-349 - this could be effectively two additional factors that influence growth on top of the four or five described in the methods. Perhaps clarify why not formally included.

Lines 408 to 410 - Note comment above - numerous other influences can complicate this result and statement. Perhaps tone down.

Lines 411 to to 421 introduce various biological management options which would vary in feasibility from place to place. Perhaps a sentence or two on feasibility and logistical considerations.

Lines 424 to 427 - correct. Poaching was speculated to influence indirectly on dynamics of black rhino in Kruger.

Author's Response to Decision Letter for (RSPB-2022-0075.R0)

See Appendix C.

Decision letter (RSPB-2022-0075.R1)

14-Mar-2022

Dear Dr Harvey Sky

I am pleased to inform you that your manuscript RSPB-2022-0075.R1 entitled "Female reproductive skew exacerbates the extinction risk from poaching in the eastern black rhino" has been accepted for publication in Proceedings B.

The AE recommended publication, but also suggest some minor revisions to your manuscript. Therefore, I invite you to respond to their comments and revise your manuscript. Because the schedule for publication is very tight, it is a condition of publication that you submit the revised version of your manuscript within 7 days. If you do not think you will be able to meet this date please let us know.

[http://datadryad.org/submit?journalID=RSPB&manu=\(Document not available\)](http://datadryad.org/submit?journalID=RSPB&manu=(Document%20not%20available)) which will take you to your unique entry in the Dryad repository. If you have already submitted your data to dryad you can make any necessary revisions to your dataset by following the above link. Please see <https://royalsociety.org/journals/ethics-policies/data-sharing-mining/> for more details.

Sincerely,

Dr Locke Rowe

Associate Editor:

Board Member

Comments to Author:

You have done a good job at introducing the changes suggested by me and the referee. Overall, this version represent a great improvement over the revised one. Nevertheless, there are a couple of minor comments from my part, including:

-lines 172-173: is there any reference that could be cited here?

-lines 412-413: Here you could mention what the extinction probabilities are for each population under a 1% level of poaching.

Author's Response to Decision Letter for (RSPB-2022-0075.R0)

See Appendix D.

Decision letter (RSPB-2022-0075.R2)

21-Mar-2022

Dear Dr Harvey Sky

I am pleased to inform you that your manuscript entitled "Female reproductive skew exacerbates the extinction risk from poaching in the eastern black rhino" has been accepted for publication in Proceedings B.

If you are likely to be away from e-mail contact please let us know. Due to rapid publication and an extremely tight schedule, if comments are not received, we may publish the paper as it stands. If you have any queries regarding the production of your final article or the publication date please contact procb_proofs@royalsociety.org

Data Accessibility section

Open Access

Paper charges

Sincerely,

Proceedings B

Appendix A

Substantive Issues

I list some points that I think require clarification.

Line 57: Strictly speaking, it is the contribution of demographic stochasticity to stochastic variance in population size (N) growth rate (λ) that depends inversely on N (Lande et al. 2003 (1.3), so that the importance of demographic stochasticity decreases as population size increases. The demographic variance, which includes both demographic stochasticity and individual heterogeneity (see next paragraph), may depend on N itself (Lande et al. 2003:19), which could alter the contribution to stochastic variance in λ .

Lines 60 – 65: I agree that reproductive skew is an important issue, but my understanding appears to be a little different from what you say here. I understand Melbourne and Hastings to have drawn a distinction between demographic stochasticity (i.e., interpreting a vital rate as a random variable applied iid to each individual, rather than applying the rate non-probabilistically to the population, or stage/age class, as a whole) and individual heterogeneity in vital rates. One can have either one without the other, e.g., it is only practicality that limits structuring in deterministic population models to stages or age classes rather than allowing complete individual heterogeneity. Fox and Kendall (2002; cited in Kendall and Fox 2003) showed that the variance, across the population, in survival when survival is structured is less than when unstructured, so structured survival can reduce extinction risk. Robert et al. showed that the opposite effect may occur for fecundity. Kendal and Fox (2003) extended their own treatment, confirming their earlier result on survival but showing that for fecundity the result depended on the details of how the vital rate was structured.

So, your account here does not appear to reflect the cited literature as I understand it, although your conclusion that individual heterogeneity in fecundity is an important issue is certainly correct. According to Kendall and Fox (2003), in what way reproductive skew effects extinction risk will depend on how that skew is structured.

Line 172: Do the sample sizes justify using AIC rather than AIC_c? Why choose $k = 7$ for the breeding GAMM when $k = 9$ gives a lower AIC score and 9 was used for the mortality GAMM?

Lines 188 – 189: The text states that at each time step, a female was first subjected to survival/mortality and then to the probability of giving birth (which makes sense to me as a new born calf would not survive its mother's death in the year of its birth) but in S2, line 16, you appear to state the opposite! The diagram in S2 does not make clear in what order potential mortality and birth were applied to an individual.

Lines 214 – 222: The issue of what regulates rhino numbers is still very much open to debate. In addition to the two papers cited [39, 40], Brodie JF, Muntifering J, Hearn M, Loutit B, Loutit R, Brell B, Uri-Khob S, Leader-Williams N, du Preez P (2011) Population recovery of black rhinoceros in north-west Namibia following poaching. *Anim Conserv* 14:354–362, also found no evidence of temporal variation in vital rates; while Ferreira SM, Greaver CC, Knight, MH (2011) Assessing the population performance of the black rhinoceros in Kruger National Park. *S Afr J Wildl Res* 41:192–204 suggested that sub-adult mortality was socially mediated, which may be an important factor in density-driven processes.

Lines 236 – 240: For clarity, perhaps you could say, ‘If the population was a proportion x of ECC above 0.75, the quantity $4(x - 0.75)$ was employed to adjust the distributions from which environmental stochasticity was drawn. The mean of...’ I gather from the R code, this quantity is what was used to scale for the density dependence.

Line 256: The description of the reproductive value factor applied to a female at a birth event is a bit vague given its importance. S2 refers to selecting the value from a Poisson distribution derived from the data. The number of yearlings raised divided by the age of the female minus 5 gives a score for each female over the age of nine. So you have a distribution of such scores. This distribution was mean-centred on zero; but also scaled? If so how (using the R function ‘scale’)? By drawing a female’s reproductive value from a Poisson distribution do you mean drawing from the ‘scaled’ empirical distribution or something more? It’s great that you have provided the R code for everything, but a reader shouldn’t have to work through the R code to understand the how the model works, as opposed to how it is implemented.

Lines 280 – 288: Are you computing the ‘stochastic growth rate’? You appear to say that you take the mean population size across realizations and compute the growth rate of this mean population size. But that is not the ‘stochastic growth rate’ of a stochastic matrix model (see Caswell [13]), which is estimated, e.g., as in Caswell (14.61).

Line 291: The demographic potential rate excludes not only stochasticity (of both kinds) and reproductive skew, but also density dependence, right?

Line 376 - 393: The three reserves were not distinguished by ECC category but only size, with Ol Jogi being the smallest (marginally). But Ol Jogi also has the poorest mean vital rates (S5), which per Table 1 is not explained by stochastic effects alone. Proximity of Ol Jogi to the other reserves does not suggest that variation in climate likely explains this difference. The reproductive skew was computed across the combined populations, but presumably the reproductive skews in S6 could be separated by reserve to see if the existing skew is greater in, e.g., Ol Jogi? If so, that would be a point of interest to the managers there. Given your lengthy consideration of other possible explanations, it seems odd that you don’t break the reproductive data up by reserve here. Even if there is no difference amongst reserves, that helps narrow the focus on the possible explanations of the poorer performance at Ol Jogi. Similarly, it would be of interest to have graphs of the age-dependent vital rates as in Figure S5.1 for the reserves to better understand the differences in means displayed in Figure S5.2.

Line 397: ‘lower heterozygosity has been linked to male reproductive success’ read literally is the opposite of Cain et al.’s findings, is it not?

Line 408: in regard to [63], don’t Cain et al. argue the opposite: that higher heterozygosity of the reproductively more successful males may compensate for their dominance in breeding.

Minor Points

Line 66: repetition of the word ‘that’

Line 96: You could also cite [39] and 'Law, P. R., Fike, B. and Lent, P. C. 2013 Mortality and female fecundity in an expanding black rhinoceros (*Diceros bicornis minor*) population. *European Journal of Wildlife Research* 59(4), 477–485. DOI: 10.1007/s10344-013-0694-y. in support of such variation too.

Line: 198: I assume you are computing, for each reserve and for each vital rate, the annual rate for each year and the standard deviation across years (consistent with Lande et al.'s 2003 (1.3) measure of environmental stochasticity), so you should say, e.g., 'annual mortality rates...' or 'annual mortality rate for each year', otherwise it looks, at first glance, like you are computing a single rate for each population, for which there's no variation.

Line 246: I don't think reference [39] states anything in support of 'Generally black rhinos reach sexual maturity around the age of 5'. The second reference given above in comment on Line 46 includes data on age at first reproduction (AFR) for that study population, with a mean of 80 months and SD of 14 months, including the fact that 13% of first births occurred to females not year of age 5. 'Sexual maturity' may well be distinct to AFR. Owen-Smith (1988: Fig 10.1) indicates an age at first conception for black rhino at a little over 5 years but earlier (p. 139) states that females attain adult status after the birth of their first calf, though sexual activity obviously predates this event. The point is citing the best source for your argument., not to quibble over your choice of age five as a cut of for reproduction.

Line 262: 'calculated' rather than 'calculate'?

Line 283: ' t and year t_{+1} ' should be ' t and year $t+1$ '.

Line 367: the apostrophe on 'habitats' seems anomalous while 'be' is missing before the final word of the sentence.

Line 589: [51] lacks source information.

Line 439: 'and encouraging that as many females as possible are breeding' is awkward wording. Perhaps 'and create conditions in which as many females as possible are breeding.'

Appendix B

Associate Editor

Comments to Author:

Two referees reviewed your manuscript and feel that it could constitute a valuable contribution but they request clarification on a range of very important issues. In particular, your interpretation of what 'reproductive skew' is lacks clarity. Referee 1 makes a very important distinction between individual heterogeneity in vital rates across individuals (demographic stochasticity) and heterogeneity in vital rates among age/stage classes, which can be incorporated in purely deterministic models. Referee 2 seems to assume that your analysis considers reproductive skew as one of the many stochastic risks faced by small populations. Your analyses first estimate age-specific vital rates and then use them to implement the projection model using an individual-based approach. Thus, in your case, it would seem as if reproductive skew is directly determined by the structured fecundity of your model and does not amount to a stochastic risk. This is an important observation as one could think of reproductive skew as a purely stochastic process whereby a fraction of individuals in the population (which each individual chosen randomly) can contribute the most to reproduction regardless of their age/stage.

We thank the editor and reviewers for their constructive comments on the manuscript, which we believe have greatly improved it. As set out in these responses, a key change to the manuscript is on our discussion of reproductive skew. Specifically, we have now explicitly defined reproductive skew as demographic heterogeneity. In other words, intrinsic variation in vital rates between individuals rather than age or stage-based variation.

See responses to Referee 1: lines 60-65 and Referee 2: comments 1-3, 5 and 11. References to R functions used added.

Another very important source of confusion is the lack of a clear presentation of the assumptions of your projection model. In particular, referee 1 rightly points out in the main text, you indicate that the order of events is survival-> reproduction but in S2 you state the opposite is true. Overall, your presentation of the projection model is rather poor. It would help to include a figure with a flow chart showing the sequence of events that individuals undergo and the underlying distribution from which probabilities of survival and reproduction are drawn.

We have corrected inconsistencies. Additionally, the flow chart in Supplementary Information S2 has been modified. See responses to referee 1: lines 188-189, lines 236-240, line 256, lines 280-288 and line 291 and referee 2: comment 7 and 8.

Both referees provide many other important comments that need to be carefully taken into consideration by either incorporating them into the manuscript or providing a very clear explanation of why you didn't do so.

Referee 1:

Line 57: Strictly speaking, it is the contribution of demographic stochasticity to stochastic variance in population size (N) growth rate (λ) that depends inversely on N (Lande et al. 2003 (1.3)), so that the importance of demographic stochasticity decreases as population size increases.

The demographic variance, which includes both demographic stochasticity and individual heterogeneity (see next paragraph), may depend on N itself (Lande et al. 2003:19), which could alter the contribution to stochastic variance in λ .

Following this point, we very much agree with the reviewer's description. The existing description has been removed as part of our reframing away from demographic stochasticity and towards demographic heterogeneity. We have now included a statement in the first paragraph that both demographic and environmental stochasticity become more important at smaller population sizes, lines 50-53.

Lines 60 – 65: I agree that reproductive skew is an important issue, but my understanding appears to be a little different from what you say here. I understand Melbourne and Hastings to have drawn a distinction between demographic stochasticity (i.e., interpreting a vital rate as a random variable applied iid to each individual, rather than applying the rate non-probabilistically to the population, or stage/age class, as a whole) and individual heterogeneity in vital rates. One can have either one without the other, e.g., it is only practicality that limits structuring in deterministic population models to stages or age classes rather than allowing complete individual heterogeneity. Fox and Kendall (2002; cited in Kendall and Fox 2003) showed that the variance, across the population, in survival when survival is structured is less than when unstructured, so structured survival can reduce extinction risk. Robert et al. showed that the opposite effect may occur for fecundity. Kendall and Fox (2003) extended their own treatment, confirming their earlier result on survival but showing that for fecundity the result depended on the details of how the vital rate was structured. So, your account here does not appear to reflect the cited literature as I understand it, although your conclusion that individual heterogeneity in fecundity is an important issue is certainly correct. According to Kendall and Fox (2003), in what way reproductive skew effects extinction risk will depend on how that skew is structured.

Thank you, this is a really helpful comment. I think the confusion/lack of clarity arose due to the varying definitions of the term 'demographic stochasticity' as discussed in Kendall and Fox (2003). We have rewritten the paragraph in question, lines 56-78. We now use the categories of stochasticity laid out in Melbourne and Hastings (2008) to frame the work, state that the type we are studying is demographic heterogeneity, and discuss the differences between structured and unstructured variation. The reproductive skew we have included in the model, as you rightly pointed out, falls into the category of demographic heterogeneity rather than demographic stochasticity as previously stated. It is also structured rather than unstructured, as previously stated. This is because the variation in fecundity is correlated 'within individuals across time (e.g., some individuals are healthier throughout their lives; "individual heterogeneity" sensu strictu Conner & White 1999)'. Quote from Kendall and Fox (2003). This is now clearly stated in lines 77 and 121.

Line 172: Do the sample sizes justify using AIC rather than AICc? Why choose $k = 7$ for the breeding GAMM when $k = 9$ gives a lower AIC score and 9 was used for the mortality GAMM?

For each model, $n=120$. Each datapoint is an age 0-40 on each of the three reserves, minus the ages 38-40 on OI Pejeta as no females survived to these ages. Number of parameters (p) in each model is 2. Formula for AICc:

$$AIC_c = AIC + \frac{2p(p+1)}{n-p-1},$$

Correction term = $2*2(2+1)/120-2-1 = 0.1025641$

Using AICc instead of AIC therefore makes very little difference to the work, and we have stuck with AIC. In graphs below blue is AIC and orange is AICc.

All analyses have now been rerun with $k = 9$ for the breeding model. This did not affect our results very much and did not necessitate changes to the interpretation and discussion. Specific changes are outlined below:

- Changes to Supplementary Information S1. Figures S1.3, S1.4, S1.5, S1.6 are new.
- Values of r and r_{long} changed in table 1
- Elasticity graphs changed in Supplementary Information S4
- Calculation of long-term population growth rate graph changed in Supplementary Information S3
- Figure 2a and 2b changed

- Projection graphs in Supplementary Information S7 changed
- Figure 3 changed
- Ages at which reproduction peaked slightly changed, line 289

Lines 188 – 189: The text states that at each time step, a female was first subjected to survival/mortality and then to the probability of giving birth (which makes sense to me as a new born calf would not survive its mother's death in the year of its birth) but in S2, line 16, you appear to state the opposite!

Apologies for the typo. The model does subject individuals to mortality first, then breeding afterwards. Lines 15-17 in S2 have been changed.

The diagram in S2 does not make clear in what order potential mortality and birth were applied to an individual.

The diagram has been modified to make this clear.

Lines 214 – 222: The issue of what regulates rhino numbers is still very much open to debate. In addition to the two papers cited [39, 40], Brodie JF, Muntifering J, Hearn M, Loutit B, Loutit R, Brell B, Uri-Khob S, Leader-Williams N, du Preez P (2011) Population recovery of black rhinoceros in north-west Namibia following poaching. *Anim Conserv* 14:354–362, also found no evidence of temporal variation in vital rates; while Ferreira SM, Greaver CC, Knight, MH (2011) Assessing the population performance of the black rhinoceros in Kruger National Park. *S Afr J Wildl Res* 41:192–204 suggested that sub-adult mortality was socially mediated, which may be an important factor in density-driven processes.

Discussion of this debate has been expanded in lines 199-210. Reference to Ferreira, Greaver & Knight (2011) added as well as Linklater & Hutcheson (2010)

Lines 236 – 240: For clarity, perhaps you could say, 'If the population was a proportion x of ECC above 0.75, the quantity $4(x - 0.75)$ was employed to adjust the distributions from which environmental stochasticity was drawn. The mean of...' I gather from the R code, this quantity is what was used to scale for the density dependence.

This has been used to describe the changes in both standard deviation and mean of the distributions in lines 216-227 and in S2 lines 62-68

Line 256: The description of the reproductive value factor applied to a female at a birth event is a bit vague given its importance. S2 refers to selecting the value from a Poisson distribution derived from the data. The number of yearlings raised divided by the age of the female minus 5 gives a score for each female over the age of nine. So you have a distribution of such scores. This distribution was mean centred on zero; but also scaled? If so how (using the R function 'scale')? By drawing a female's reproductive value from a Poisson distribution do you mean drawing from the 'scaled' empirical distribution or something more? It's great that you have provided the R code for everything, but a reader shouldn't have to work through the R code to understand the how the model works, as opposed to how it is implemented.

Details on how this was done have now been added to lines and 245-255 and Supplementary Information S2, lines 34-44. Furthermore, for clarity in the terminology used and to prevent confusion with previous population ecology literature, we have changed the 'reproductive value' to a 'reproductive score' as 'reproductive value' has a different definition in Caswell (1978).

Lines 280 – 288: Are you computing the ‘stochastic growth rate’? You appear to say that you take the mean population size across realizations and compute the growth rate of this mean population size. But that is not the ‘stochastic growth rate’ of a stochastic matrix model (see Caswell [13]), which is estimated, e.g., as in Caswell (14.61).

Some of the description of how this was done has been moved to Supplementary Information S3 due to constraints on word count. This term has been removed, and the description of r and r_{long} restructured and description of r_{long} changed in lines 273-282, the description of table 1 and in Supplementary Information S3 lines 3-18. It is now termed ‘long-term realised population growth rate’.

Line 291: The demographic potential rate excludes not only stochasticity (of both kinds) and reproductive skew, but also density dependence, right?

Yes. Clarified in line 277-280

Line 376 - 393: The three reserves were not distinguished by ECC category but only size, with OI Jogi being the smallest (marginally). But OI Jogi also has the poorest mean vital rates (S5), which per Table 1 is not explained by stochastic effects alone. Proximity of OI Jogi to the other reserves does not suggest that variation in climate likely explains this difference. The reproductive skew was computed across the combined populations, but presumably the reproductive skews in S6 could be separated by reserve to see if the existing skew is greater in, e.g., OI Jogi? If so, that would be a point of interest to the managers there. Given your lengthy consideration of other possible explanations, it seems odd that you don’t break the reproductive data up by reserve here. Even if there is no difference amongst reserves, that helps narrow the focus on the possible explanations of the poorer performance at OI Jogi. Similarly, it would be of interest to have graphs of the age-dependent vital rates as in Figure S5.1 for the reserves to better understand the differences in means displayed in Figure S5.2.

You are correct that the variation in climate between the three reserves is not very big, and the mention of seasonality has been removed. However, the community of plants present on each reserve is very different (Adcock K, Amin R, Khayale C. 2007 Habitat characteristics and carrying capacity relationships of 9 Kenyan black rhino areas.). We are also conducting other work that looks at black rhino diets, and it suggests that black rhino diets on OI Jogi are less valuable than the other two reserves. If true, this may lead to suggestions that the ECC categories that you reference in your comment may actually need to be rethought, and OI Jogi may fall into a different category than the other population. We therefore do not want to completely get rid of the discussion suggesting that habitat differences may lead to population performance and risk of extinction.

The suggested graphs have been added to Supplementary Information 5 and 6, and referred to in the results in lines 291-293, and 357-359. Sentences saying that reproductive skew is unlikely to cause OI Jogi to have poorer performance lines 355-359.

Line 397: ‘lower heterozygosity has been linked to male reproductive success’ read literally is the opposite of Cain et al.’s findings, is it not?

Amended and worded more accurately line 366-367.

Line 408: in regard to [63], don’t Cain et al. argue the opposite: that higher heterozygosity of the

reproductively more successful males may compensate for their dominance in breeding.

Noted in the text lines 382-384.

Minor Points

Line 66: repetition of the word 'that'

One removed

Line 96: You could also cite [39] and 'Law, P. R., Fike, B. and Lent, P. C. 2013 Mortality and female fecundity in an expanding black rhinoceros (*Diceros bicornis minor*) population. *European Journal of Wildlife Research* 59(4), 477–485. DOI: 10.1007/s10344-013-0694-y. in support of such variation too.

Has been added

Line: 198: I assume you are computing, for each reserve and for each vital rate, the annual rate for each year and the standard deviation across years (consistent with Lande et al.'s 2003 (1.3) measure of environmental stochasticity), so you should say, e.g., 'annual mortality rates...' or 'annual mortality rate for each year', otherwise it looks, at first glance, like you are computing a single rate for each population, for which there's no variation.

Wording changed to clarify lines 187-191

Line 246: I don't think reference [39] states anything in support of 'Generally black rhinos reach sexual maturity around the age of 5'. The second reference given above in comment on Line 46 includes data on age at first reproduction (AFR) for that study population, with a mean of 80 months and SD of 14 months, including the fact that 13% of first births occurred to females not year of age 5. 'Sexual maturity' may well be distinct to AFR. Owen-Smith (1988: Fig 10.1) indicates an age at first conception for black rhino at a little over 5 years but earlier (p. 139) states that females attain adult status after the birth of their first calf, though sexual activity obviously predates this event. The point is citing the best source for your argument., not to quibble over your choice of age five as a cut of for reproduction.

Wording and references changed to make this more accurate lines 230-232

Line 262: 'calculated' rather than 'calculate'?

Changed

Line 283: ' t and year $t+1$ ' should be ' t and year $t+1$ '.

Changed

Line 367: the apostrophe on 'habitats' seems anomalous while 'be' is missing before the final word of the sentence.

The apostrophe was to indicate it was a quote from the reference. This has been changed to " to make this more clear. The word 'be' has been added where it was missing

Line 589: [51] lacks source information.

Has been added.

Line 439: 'and encouraging that as many females as possible are breeding' is awkward wording. Perhaps 'and create conditions in which as many females as possible are breeding.'

Changed to 'encourage as many females as possible to breed' line 414

Referee 2:

Comments:

1) Considering reproductive skew is important for eastern black rhino given that most populations are small and would be vulnerable to stochastic processes. The manuscript could benefit from perhaps highlighting stochastic risk associated with the persistence of small populations. Reproductive skew is one of those stochastic risks that realize whether a species is social or not. This could compliment the existing rationale.

This is a great point. As is also stated in our responses to the previous reviewer above, the first version of the manuscript lacked clarity on the specific definition and description for reproductive skew. Following reviewer 1s suggestion, we now frame reproductive skew as an intrinsic form of demographic heterogeneity, rather than a stochastic risk. We now explore this in the introduction lines 56-79. New description of stochastic factors that affect populations, especially small ones, in lines 50-55.

2) One aspect that would improve the manuscript is a more explicit definition of reproductive skew. The present definition provided in lines 72-73 is confusing – that definitions comprise various aspects of vital rates (birth rate, survival during the suckling period etc.). This contrast an understanding that reproductive skew is a partitioning of reproduction among the same sex.

New definition offered in lines 66-67. Old definition removed.

3) A third element of clarification is to provide some prediction of what measures can reflect reproductive skew that makes it different from simple variability in reproductive output as indicated that associate with age, environmental drivers (e.g. see papers on drought effects on birth rates), synchronization of reproduction and variability within an individual cow.

Addressed in lines 58-65 and 68-70.

4) The authors could add additional value in the discussion on the possible consequences of reproductive skew for meta-population dynamics, particularly if future rhino management seek actively to mimic meta-population processes. For instance, Eastern black rhino occur in relatively small-protected areas that reflects patches of suitable habitat embedded in a landscape of unsuitable habitat. A species may persist locally and evolutionary if they overcome the risks imposed by landscape fragmentation of suitable habitat. In such instances populations, albeit small, that occasionally interact with other populations create a meta-population that has, as a whole, a higher chance of persistence than individual populations on their own. Meta-populations typically (1) span landscapes that have patches of suitable habitat supporting local breeding populations, with vacant habitat patches that individuals can colonize; (2) suitable habitat vary on those patches resulting in different birth and death rates in asynchrony between patches; (3) colonization and extinction of species on a patch takes place; and (4) dispersal occurs between local populations occurring on patches. For large mammals, the time- and spatial-scale over which population dynamics play out

results in more lenient meta-population criteria including (1) local breeding populations should be discrete rather than inhabiting discrete habitat patches; and (2) local populations should have dissimilar growth rates. The Eastern black rhinos may thus function as a meta-population within Kenya. The present analyses treated each population in isolation and defined how reproductive skew under various levels of poaching impact extinction risk of each population. Reflecting on how reproductive skew under different levels of poaching impact extinction risk of the meta-population could add additional value to the manuscript and practical implications for conservation managers.

We thank the reviewer for this comment. We now mention explicitly that Kenyan black rhinos are managed as a meta-population in line 95. We also have added discussion of metapopulations in Lines 370-379. Discussion of how metapopulation management could be used to alleviate reproductive skew if it is caused by social factors is in lines 421-423.

5) Some elements that are missing are the mechanisms that could lead and maintain reproductive skew. Allee effects of reduced mating opportunities for instance etc. This is important because the key premise of the future predictions is that the patterns of reproductive skew remains constant over a 100 year period in both the individual and matrix population models.

This is an important comment giving more context to the discussion on reproductive skew. Suggestions for the causes of skew are now included in lines 101-107. A statement and justification of the assumption that reproductive skew distribution stays constant in lines 256-259 and Supplementary Information 2 lines 34-35. Discussion of why poaching is different to other sources of mortality and may perpetuate skew because it is likely to kill fit/productive females as well as less fit/frail ones in lines 400-405. Furthermore, see response to comment 11.

6) It could be useful to have a figure or table that illustrate the observed reproductive skew in the three populations. At present, this resides in the supplementary material.

We are very close to the page limit, and to put such a figure in the manuscript, we would have to remove another section. We have added figures that show reproductive skew on each reserve to Supplementary 6 (see response to reviewer 1 Line 376 – 393). As differences in reproductive skew probably do not account for the differences in population performance, we consider these figures to be less necessary in the manuscript than the material that is currently there.

7) Some clarity is needed with regards to long term growth estimates derived from year 25 to year 100. The analyses modelled density-dependence. This means that population growth will not be constant over time. It is thus hard to interpret the meaning of the growth rate comparisons.

This comment raises a good point and highlighted the need for us to improve the clarity of our descriptions of this long-term growth rate. Firstly, as review 1 pointed out, we incorrectly described this as the stochastic growth rate (see response), which has a specific definition for structured population models. But the reviewer raises another good point, which is that because of density dependence we cannot observe an asymptotic population growth rate. Instead, what we observe is that following a period of transient dynamics owing to unstable age-structure of 25 years, populations reach a kind of stochastic equilibrium, during which population growth rates vary around an approximately stable average, which can be found in Supplementary 6. Therefore, we aimed to capture this long-term average across the simulation. To address this we have thoroughly

revised our descriptions of the long-term growth rates explored in lines 273-282 and Supplementary Information S3.

8) Some additional considerations on demographic responses as population increase could be useful that considers sequential responses of calf survival, fecundity and then adult survival (see Eberhardt, L.L., 2002. A paradigm for population analysis of long-lived vertebrates. *Ecology*, 83(10), pp.2841-2854, as well as Sibly, R.M., Barker, D., Denham, M.C., Hone, J. and Pagel, M., 2005. On the regulation of populations of mammals, birds, fish, and insects. *Science*, 309(5734), pp.607-610.). This could provide an alternative approach to estimate ECC based on the responses noted by rhinos in the data.

A consideration of these demographic responses as populations increase is presented in Supplementary Information S4. The elasticity analysis was removed from the manuscript due to limitations of space. A reference to Sibly et al. (2005) has been added to line 220. We think that the way ECC has been calculated largely agrees with the theoretical proposition of large, long-lived species having convex relationships between population size and growth rate.

9) The first part of the discussion may be best integrated into a short description of the study sites.

Apologies if we have misinterpreted this comment, but we opted to frame the start of the discussion as a general paragraph describing our key results from the study, and think this a justified way of presenting the manuscript. Discussion of reserve-level differences is included later on in the manuscript, and references to sizes and distance apart have been removed, as this is stated in the methods, lines 130-135

10) It could be useful to also add in the discussion some reflection of sensitivity of population persistence of reproductive skew in large populations under different poaching levels in combination with an element of meta-population consequences.

See response to comment 4. Discussion of why poaching is different to other sources of mortality because it is likely to kill fit/productive females as well as less fit/frail ones in lines 400-405.

11) The discussion do provide two “management recommendations”, but could benefit from including more detail on mechanisms that can lead to reproductive skew and then how to explicitly address a mechanism (see Ferreira, S.M., le Roex, N. and Greaver, C., 2019. Species-specific drought impacts on black and white rhinoceroses. *Plos one*, 14(1), p.e0209678; le Roex, N. and Ferreira, S.M., 2021. Rhino birth recovery and resilience to drought impact. *African Journal of Ecology*, <https://doi.org/10.1111/aje.12854>; le Roex, N. and Ferreira, S.M., 2020. Age structure changes indicate direct and indirect population impacts in illegally harvested black rhino. *PloS one*, 15(7), p.e0236790).

Firstly, please see response to comment 5. We have now included discussion of how management recommendations depend on the potential mechanisms that cause reproductive skew in lines 413-423. Discussion of how poaching may actually exacerbate reproductive skew, leading to a feedback loop is now in lines 424-432 with reference to le Roex & Ferreira (2020).

12) Overall, this paper has an important contribution to be made, but requires differentiation between

consequences of small populations, demographic variation and reproductive skew as a component of demographic variation.

We thank the reviewer for this positive feedback, and hope that our changes to the manuscript are more explicit in their descriptions of demographic heterogeneity, skew and its differences compared to stochasticity.

Appendix C

Associate Editor

Comments to Author:

Your resubmission was evaluated by one of the referees who reviewed your original submission. Overall, the resubmitted manuscript represents a great improvement over the original version. There are, nevertheless, some need for further clarifications as per the referee comments.

We would like to thank the editor for the time spent on this manuscript. Please see responses to the referee comments below.

From my part I would like to highlight that you seem to use “reproductive skew” and “structured demographic heterogeneity” as if they were exchangeable terms but they are not. Reproductive skew is one type of “structured demographic heterogeneity”. To avoid confusion, please remove “defined as structured demographic heterogeneity” in line 76 and replace “structured demographic heterogeneity” in line 119 with “reproductive skew”. This also applies to line 395.

We appreciate that the terms are not interchangeable, so thank you for pointing out where we have used them as such. We have made the suggested changes in those three places as well as line 329, 397, 408.

Reviewer(s)' Comments to Author:

Referee: 2

Comments to the Author(s).

Dear Editor,

I previously reviewed this paper. The authors have addressed several concerns well in the revision.

Although the focus is on black rhino, the outcomes of reproductive skew on dynamics of large mammal species carry broader importance. The benefits of large mammal species to ecosystems and people further get degraded by various global environmental change drivers. Understanding the consequences of reproductive skew provide opportunities for authorities to anticipate and respond.

We appreciate the referee's previous comments and believe the manuscript is much improved for them. We also thank the reviewer for taking the time to look at this again, and we have addressed these comments as fully as possible.

Some small comments:

Lines 106 to 114 - Perhaps reflect extinction risk at a local scale - extinction risk of a local

population of black rhino. Extinction risk should be defined here already - I imagine the authors focus on functional extinction. Perhaps clarify how PVA considers this local functional extinction risk.

Extinction is defined as a population reaching a female population of 0, although it is noted that 'populations would become functionally extinct before then' in lines 109-110

Line 123 to 125 - This is an important focus, but could also be done without using any empirical data. Perhaps the authors can clarify the use of a specific case study as they have done.

Clarification added that we are used the empirical data as a case study, line 123

Lines 140-142 - The authors make clear their assumptions. Male availability, however, could be key at low densities. Perhaps reflect the potential risk of this assumption in the discussion and dismiss it as a concern for the three case studies.

This is discussed in lines 381-384

Lines 164 to 167 - Perhaps it will add value if the authors include the definition of age-specific mortality and birth rates. For example calves (was it males and females) born per female of age x? It is not clear if this is now a summary variable disregarding reproductive skews?

Clarified in lines 164-167

Lines 187 to 195 - The authors should reflect on papers on drought responses and resilience of black rhinos to environmental perturbations - limited responses and resistance to drought suggest the authors over-emphasize the role of environmental stochasticity - perhaps reflect in the discussion. See papers by le Roex, N. and Ferreira, S.M., 2021. Rhino birth recovery and resilience to drought impact. African Journal of Ecology, 59(2), pp.544-547. and Ferreira, S.M., le Roex, N. and Greaver, C., 2019. Species-specific drought impacts on black and white rhinoceroses. Plos one, 14(1), p.e0209678.

Discussed in lines 351-356. Both suggested papers cited here.

Lines 214 to 225 - The authors should reflect on Social Carrying Capacity given the reflections on intra-specific harassment and then rationalize that the ECC approach may also reflect a social carrying capacity concept.

Social factors are discussed, and a justification for use of the term ECC is provided, in lines 203-204, line 208-210.

Lines 271 to 280 - the description highlight four variables' influence benchmarked against a best model. In effect is there a fifth variable as well which is poaching

pressure. This is a complex set of combinations and perhaps the authors could clarify in a summary table.

This table has been added in at the end of the methods. We have cut the wordcount down elsewhere to make space for this table.

Table 1 - are there any confidence intervals?

This is now table 2. r is calculated from λ , the dominant eigenvalue of Leslie matrices, so it is mathematically defined and does not have confidence intervals. We are treating this as the average demography or 'world stays the same' scenario, and the variability is captured in the full stochastic projection model that we used to calculate r_{long} (see Supplementary Information S3). As the r_{long} values are calculated from the average of 500 projections on each reserve, it is possible to calculate confidence intervals, although this has required a slight change of methods, which was in fact an improvement in its own right. Previously to calculate r_{long} , we:

- calculated the mean population size for all 500 simulations at each timestep
- used those mean values to calculate the population growth between year t and $t+1$
- calculated the average of these growth values between years 25 and 100.

When calculating the confidence intervals, we realised that there was a better way to do this. We:

- calculated the population growth rate between year t and $t+1$ for each of the 500 simulations
- calculated the mean values and 95% confidence intervals for all 500 simulations for each year
- averaged the mean, lower and upper confidence intervals between years 25 and 100

This means that the r_{long} values in table 2 have changed a little, but this does not affect the interpretation or discussion. In Supplementary Information S3, we have updated the methods in lines 11-16, changed figure S3.1, and added in new figures (S3.2) that show the r_{long} values on each reserve with the confidence intervals.

Brief mention of why the OI Jogi population may have a more variable r_{long} lines 340-341.

Figure 3 - Perhaps the authors should reflect what the level of reproductive skew was and whether this did associate with the magnitude of shift in probability of extinction as % annual off take of adults increased. One may ask, at what threshold of reproductive skew are effects negligible irrespective of demographic, environmental and density-dependence factors. But I do note some observations in Lines 354 to 357. This will have impact on the discussion in lines 391 to 406. It does appear that several other factors may influence the sensitivity of population performance to the interaction between

reproductive skew and poaching pressure. The authors will need to reflect on this. **This is addressed with the next comment.**

Lines 348-349 - this could be effectively two additional factors that influence growth on top of the four or five described in the methods. Perhaps clarify why not formally included.

Clarification added in lines 344-347 that population size, growth rates or reproductive skew were not artificially altered as this study focused on real world populations and was data-driven. Suggestion added that future work that uses simulated data could look at the importance of these factors by varying them.

Clarification the level of reproductive skew found in our study population significantly increases extinction risk on line 397, avoiding the implication that any level of skew would have the same effect on extinction risk. Acknowledgement that a key research question will be to find the threshold of female reproductive skew, which will be very useful for management, in lines 403-406.

Lines 408 to 410 - Note comment above - numerous other influences can complicate this result and statement. Perhaps tone down.

This has been reworded to reflect the nuance that you rightly mention in lines 411 - 412

Lines 411 to to 421 introduce various biological management options which would vary in feasibility from place to place. Perhaps a sentence or two on feasibility and logistical considerations.

Sentences added in lines 420-423 and 425-427

Lines 424 to 427 - correct. Poaching was speculated to influence indirectly on dynamics of black rhino in Kruger.

Great, thank you!

Female reproductive skew exacerbates the extinction risk from poaching in the eastern black rhino

Nick Harvey Sky^{ab}, John Jackson^c, Geoffrey Chege^d, Jamie Gaymer^e, David Kimiti^f, Samuel Mutisya^g,
Simon Nakito^g, Susanne Shultz^a

^a Department of Earth and Environmental Sciences, University of Manchester, Manchester, M13 9PL,
UK

^b North of England Zoological Society, Chester Zoo, Caughall Road, Chester, CH2 1LH, UK

^c Department of Zoology, University of Oxford, OX1 3SZ, UK

^d Lewa Wildlife Conservancy, P.O. Box, Private Bag, Isiolo-60300, Kenya

^e Ol Jogi Ltd., PO Box 259–10400, Nanyuki, Kenya

^f Grevy's Zebra Trust, PO Box 15351 – 00509, Nairobi, Kenya

^g Ol Pejeta Conservancy, PO Box 167, Nanyuki, Kenya

Corresponding author – Nick Harvey Sky nick.c.harvey@gmail.com

Acknowledgements

We express sincere thanks to the security, monitoring and research teams on all three reserves who meticulously collected the demographic data, especially I. Lemaiyan who was integral to this on Lewa. We would also like to thank F. Omengo and KWS for providing permission to conduct this study and to M. Kamau, A. Kibungei and B. Gituku for comments on an early version of the manuscript.

Funding

N.H. is funded by the NERC Manchester-Liverpool DTP, NE/L002469/1, and Chester Zoo on a CASE studentship

Abstract

Variation in individual demographic rates can have large consequences for populations. Female reproductive skew is an example of structured demographic heterogeneity where females have intrinsic qualities that make them more or less likely to breed. The consequences of reproductive skew for population dynamics are poorly understood in non-cooperatively breeding mammals, especially when coupled with other drivers such as poaching. We address this knowledge gap with population viability analyses using an age-specific, female-only, individual-based, stochastic population model built with long-term data for three Kenyan populations of the Critically Endangered eastern black rhino (*Diceros bicornis michaeli*). There was substantial reproductive skew, with a high proportion of females not breeding or doing so at very low rates. This had a large impact on the projected population growth rate for the smaller population on Ol Jogi. Moreover, including female reproductive skew exacerbates the effects of poaching, increasing the probability of extinction by ~70% under a simulated poaching pressure of 5% offtake per year. Tackling the effects of reproductive skew depends on whether it is mediated by habitat or social factors, with potential strategies including habitat and biological management respectively. Investigating and tackling reproductive skew in other species requires long-term, individual-level data collection.

Keywords

Population viability analysis, demographic variance, vital rates, reproductive skew, poaching, extinction risk

1. Introduction

With over 1 million species facing extinction [1] and a 68% average decline in monitored vertebrate populations between 1970-2016 [2] quantifying extinction risk and identifying its drivers in vulnerable populations is crucial for biodiversity conservation. Small populations are particularly vulnerable to extinction [3] because environmental and demographic stochasticity [4], inbreeding depression, loss of genetic diversity [5] and inverse density dependence (or Allee effects) [6], all become more important as population size decreases. Understanding the drivers of extinction and population dynamics in small populations is vital to the development of effective conservation strategies.

The causes of unequal realised vital rates (survival, growth and reproduction) across individuals that can disproportionately impact small populations can be grouped into different categories [7].

Demographic stochasticity is caused by random variation in probabilistic birth and death rates resulting in some individuals contributing more offspring or living longer by chance. Extrinsic factors

contribute to environmental stochasticity, which is spatial or temporal variation in birth and death rates at the population level. In contrast to extrinsic factors and demographic stochasticity, which lead to chance variation in realised reproductive rates between individuals, demographic heterogeneity is intrinsic variation in vital rates at the individual level, which may be due to genetic quality, maternal effects, access to resources or different exposure to stressors [8,9].

Reproductive skew, defined as unequal reproductive success between individuals of the same sex in a population [10,11], is one facet of demographic heterogeneity that may affect extinction risk differently to other variations in reproductive success. Reproductive skew is predicted to affect population growth independently of overall reproductive potential, population age structure and environmental changes [12]. Structured, as opposed to unstructured, variation in vital rates is not independent of the vital rates of other individuals or the identity of the individual [13]. It has been shown that structured demographic heterogeneity that is retained throughout individuals' lifetimes, or 'individual heterogeneity' can have significant effects on extinction risk in small populations [14]. Despite this, population models often incorporate demographic stochasticity, in which all individuals have equal vital rates but the chance of a demographic event is modified by sampling variance, but do not include demographic heterogeneity [15]. Disentangling age-specific reproduction from female reproductive skew and assessing their impacts on population viability will add important dynamics of individual heterogeneity into demographic studies.

For mammals there is often an assumption that, outside of cooperatively breeding species, there is little reproductive skew among females or it is not important [16,17]. Long-term datasets have shown that there is significant female reproductive skew in non-cooperatively breeding mammals [18,19] but few studies have investigated its effect on population viability.

The effects of female reproductive skew may be magnified when coupled with other drivers of population dynamics such as poaching. Poaching has been shown to increase male reproductive skew in African elephants (*Loxodonta africana*) [20] and catastrophic poaching of male saiga antelopes (*Saiga tatarica tatarica*) led to a crash in the number of pregnancies due to disturbed mating behaviour [21]. The combined effect of female reproductive skew and poaching on population viability has yet to be explored. If population persistence depends on the survival and reproduction of a small number of fecund individuals, then the death of any of those individuals due to poaching may drastically increase extinction risk.

One Critically Endangered subspecies for which assessing extinction risk is vital is the eastern black rhino (*Diceros bicornis michaeli*) as poaching is still a major threat[22]. Kenya, the major range state for the subspecies, has set a target of 'net growth of at least 5% per annum maintained in at least six

established populations' [23], which some reserves are not achieving [24]. To minimise poaching risk, Kenyan black rhinos are managed as an artificial meta-population [23]. Isolated populations present the opportunity to conduct proxy natural experiments for studying environmental and demographic drivers of population dynamics. Reproductive skew has been identified within both captive and free-living female black rhinos, including variation in the number of calves, age of first reproduction and inter-calving interval [24–26]. Intrinsic differences in quality may allow particular individuals to benefit from both environmental and social factors. Females are largely solitary, but regularly interact with other males and females which have adjacent or overlapping home ranges [27,28]. Dominant females may secure home ranges with better quality diets and breed more successfully. Intraspecific aggression, harassment from males or other females [29,30] and fighting [12], could also cause stress and inhibit reproduction [18], with fitter females more likely to successfully harass others while resisting it themselves.

The aim of our study was to assess the local extinction risk of eastern black rhino populations in three Kenyan reserves under multiple drivers of population dynamics including poaching and reproductive skew. We constructed a population viability analysis (PVA), a method which is routinely used to assess the risk of extinction faced by a species or population over a particular time period [31,32] and the importance of particular threats [33]. We defined local extinction as a female population of zero, although populations would become functionally extinct before then. The utility of PVAs to effectively quantify extinction risk and predict future population declines is reliant on parameterisation from high quality data and appropriate life-history assumptions[34]. Black rhinos in Kenya are intensively monitored as part of efforts to protect them from poaching and so there are excellent current and historical demographic data available.

We used a data-driven approach to estimate population viability using approximately 40 years of individual-based demographic data. Such long-term datasets are very rare, particularly for free-living populations of a Critically Endangered species. To explore extinction risk, we constructed an age-specific, female-only, individual-based, stochastic population model. An individual-based approach allowed us to include reproductive skew as an intrinsic reproductive score that was relative to the entire population, assigned to each individual at birth and stayed the same through their entire lives. As well as reproductive skew, the model incorporated density dependence, environmental stochasticity and demographic stochasticity. We then simulated population growth over the next 100 years for each reserve. Crucially, we used this as a case study to simulate population growth under different offtake scenarios to assess the effect that reproductive skew and different levels of poaching would have on population viability for a large mammal.

1. Methods

2.1 Study populations

We focused on three different Kenyan reserves. The 250km² Lewa Wildlife Conservancy in Meru County (0.20°N, 37.42°E) founded a 20km² rhino sanctuary in 1983 and converted completely to a conservancy in 1995. The 360km² Ol Pejeta Conservancy in Laikipia Country (0.02°N, 36.90°E) founded a rhino sanctuary in 1988. The 235km² Ol Jogi Conservancy in Laikipia County (0.32°N, 36.98°E) was established as a rhino sanctuary in 1980.

2.2 Data collection

Due to intensive monitoring to protect from poaching, there are high quality individual-based demographic data available. Black rhino calves usually stay with their mothers for around 2.5-3 years, making the assignment of maternity almost certain. Paternity is difficult to assign without the use of genetic techniques [35] and we could not include males in this study. High skew in male breeding success [36] suggests a lot of extra breeding potential among males, so generally breeding is unlikely to be limited by male density. In mammalian population modelling, 'female dominance' is often assumed where there are always enough males to fertilise all females [37]. A female-only design was therefore appropriate.

The three reserves record the dates of births, deaths, imports (including the ages of imported individuals) and exports. The ages at which females died and gave birth has been accurately recorded since the foundation of each sanctuary. Data are available for the periods 1984-2019 for Lewa, 1980-2019 for Ol Jogi and 1990-2019 for Ol Pejeta.

We constructed a time-to-event demographic dataset for each population documenting whether each female died, bred, or was imported/exported in a given year since they were present in the population. For import and export events, individuals were brought in or removed from the population with no birth or death event. Individuals were only ever translocated once in their lifetimes. 10 females were imported to and 10 exported from Lewa, 19 to and 2 from Ol Jogi and 18 to and 2 from Ol Pejeta. Imported and non-imported individuals were not differentiated but we incorporated individual-level differences which capture some of the potential for variation between imported and non-imported individuals. The demographic outcomes were coded as binary response variables, where a 1 indicated birth or survival in a given observation year. 34 calves died before their sex was recorded, 8 on Lewa, 25 on Ol Pejeta and three on Ol Jogi. We randomly selected whether each of these was a male or a female and removed the simulated male calves from the

model. After this there were 99 females recorded from Ol Pejeta, 79 from Lewa, 55 from Ol Jogi and a total of 2252 year-age observations.

2.3 Estimation of age-specific vital rates

We constructed an age-specific model that incorporated vital rate changes across lifespan, an approach recently applied to Asian elephants (*Elephas maximus*) [38]. All analyses were done in R version 4.0.1 [39].

Mortality and birth events in the demographic records were used to quantify population vital rates. The proportion of females that died or gave birth to female calves at each specific age provided raw age-specific mortality and birth rates that did not include reproductive skew. The values used in the PVA were estimated from the raw data using generalised additive models (GAMs) implemented in the *mgcv* package [40]. The distributional assumptions were checked using the *DHARMA* package [41] (Supplementary Information 1).

2.4 Parameterisation of stochastic projection model

To assess the future viability of the three populations, we built female-only, stochastic individual-based projection models using the predicted age-specific vital rates and projected 100 years into the future. Projections started from the populations present at the end of 2019. We cannot present the age structures of these starting populations due to confidentiality of black rhino data. An individual-based modelling framework allowed us to incorporate demographic stochasticity, an important source of uncertainty and an advantage over deterministic models. Every year, whether an individual died or bred was simulated using a Bernoulli distribution determined by the probabilities calculated using the GAMs for each reserve (a single trial for each living individual in each year) using the `rbinom()` function. Reproduction was dependent on survival, as it occurred after survival/death. We removed individuals over the age of 40 in each year of each simulation (probability of mortality for individuals aged 40 was given a value of 1), because few individuals survived over this age and there was large variation in life-history parameters. No individuals over the age of 40 reproduced.

2.5 Estimate of environmental stochasticity

We estimated environmental stochasticity from observed variance in annual vital rates across the study period. We calculated the annual mortality rate for the whole population on each reserve, and the annual birth rate for reproductive ages (5 to 34 years of age) from foundation to 2019. We then calculated the standard deviation of each of these annual vital rates, which represented the environmental stochasticity for each vital rate in each population. To incorporate these into the projections, every year of the simulation we sampled from two truncated normal distributions

created using `rtruncnorm()`. These had a mean of zero, a standard deviation equivalent to that of the annual vital rates and were truncated at 0.5 and -0.5 to prevent unrealistic jumps in population size. The breeding and mortality probability of all individuals were then modified separately every year by these simulated factors. After this we ensured that no individuals below the age of 5 bred, as it was assumed to be a pre-reproductive life stage.

2.6 Estimate of density dependence

Whilst there is evidence for declines in reproduction when black rhinos increase above habitat-specific densities [28], it is uncertain what density-dependent factors regulate their numbers [42,43]. In variable environments like African savanna carrying capacity is dependent on resource availability [44]. Increasing densities of rhinos can reduce diet availability through browsing pressure [45] but it primarily depends on rainfall [46]. The concept of a fixed ecological carrying capacity (ECC) is therefore not particularly meaningful in areas with variable rainfall density [47]. Intra- and intersexual competition may also be important in density-driven variations in vital rates [48]. Male harassment of females can decrease recruitment rates, and variation in fecundity can be driven by sex-ratio and density [30]. Social carrying capacity may be a more accurate description of a maximum density of rhinos than ECC. Regardless of the mechanism, black rhino populations cannot grow in a limited area indefinitely, although pre-20th century maximum population densities are unknown. Therefore, there needs to be a way of including density dependence in a biologically relevant way.

Carrying capacity was estimated using the IUCN translocation guidelines which incorporate reserve size, average annual rainfall and habitat type [28]. We termed it ECC to be consistent with the IUCN even though it may be determined by both environmental and social factors. All three reserves fall into the category '0.2-0.4 rhino per km²'. We assumed that maximum density is 0.4km², which gives ECC estimations of 100 for Lewa, 140 for Ol Pejeta and 90 for Ol Jogi. Assuming equal sex ratios, these were halved to give the predicted female ECCs.

We incorporated density dependence into the stochastic projection models using the hypotheses that density dependence only has a significant effect above 0.75(ECC) [28], and populations can increase above the estimated ECC [49]. We also mediated density dependence using environmental stochasticity. Theoretically, large, long-lived species exhibit convex relationships between population size and growth rate [50]. Below 0.75(ECC), environmental stochasticity was calculated as in section 2.5. If the population was a proportion x above 0.75(ECC), then the distributions from which environmental stochasticity was drawn were altered using $4(x - 0.75)$. The standard deviations were increased by adding $4(x - 0.75)$ to the standard deviation calculated in section 2.5. The mean of the

sampling distributions for breeding and mortality probabilities were decreased or increased from zero by $4(x-0.75)$ respectively. The distributions were truncated at -0.5 and 0.5 to prevent biologically unrealistic jumps in population size .

2.7 Reproductive skew

We conservatively estimated female reproductive skew by calculating the number of calves each female over the age of 9 had successfully raised to the age of one year. Generally, the earliest that black rhino females can calve is 5 years old [51], although some females have been recorded to calve between the ages of 4 and 5 [26]. The number of nulliparous females was dominated by younger individuals, and we considered that including all females over the age of 5 would inflate the reproductive skew present in the population. We chose 9 years as the average age of first calving is around 7 [51] and 9 is around the first peak of reproductive probability. The number of yearlings each female had produced was divided by their age above 5, to give an annual rate of yearling production.

We calculated a reproductive score using both male and female calves as part of efforts to estimate it conservatively. With a 50:50 sex ratio, the skew would be the same as with just female calves. However, due to the relatively small sample size, using only female calves increased estimates of reproductive skew. This distribution was only used to estimate the reproductive score of females relative to each other, and resulting values were scaled to conserve the average breeding probabilities calculated using the female-only vital rates. Males therefore do not feature in the model.

The distribution of reproductive success was created using `hist()`. At the start of the projection, or at birth, each female was assigned a relative reproductive score, that stayed with them throughout their lives, using the distribution of the rate of yearling production. Firstly, a value was drawn from this distribution using `sample()`, assigning each individual an integer of one to ten, according to the probabilities from the distribution. All reproductive scores of new individuals were scaled around zero using `scale()`, to preserve the average breeding probability of each population. Reproductive values were divided by 100 so that the final highest modifications were an order of magnitude lower than the annual breeding probabilities of reproductive age females, which were between 0.1 and 0.2. Every year of the simulation, the probability that each individual female reproduced was altered by their relative reproductive score (Supplementary Information 2).

We assumed that the reproductive skew distribution stayed constant. If female reproductive skew is caused by social interactions such as harassment, or Allee effects [52], then density changes may

affect it. However, as we could not attribute definite causes to reproductive variance, we could not confidently alter it as the social context changed.

A formal description of the model, following the protocol set out in Grimm et al. (2006) [53], can be found in the Supplementary Information (S2).

2.8 Analysis

2.8.1 Reproductive skew and poaching

We projected the change in population sizes with and without reproductive skew to test the impact that female reproductive skew has on each population under different poaching regimes. We used a Bernoulli distribution created with `rbinom()` to decide whether each adult over the age of 5 would be poached in a particular year. We refer to this as the percentage annual adult offtake. We increased offtake from 0% to 20%, which allowed us to compare how the probability of extirpation of each population over 100 years changed with and without reproductive skew under different simulated poaching regimes.

2.8.2 Long-term growth rates

In order to explore the effect of stochasticity, reproductive skew and density dependence on the projected growth rates, we first estimated the intrinsic rate of increase r of each population using Leslie matrices [37], which we term the demographic potential growth rate. We then explored the difference between demographic potential growth rates and long-run realised population growth rates accounting for environmental stochasticity, demographic stochasticity, density dependence and reproductive skew by calculating the 'long-term annual population growth rate' r_{long} using our stochastic projections. Full methods and mathematical justifications can be found in Supplementary Information S3.

Table 1 shows a summary of the variables included in each analysis. Supplementary Information S4 presents an elasticity analysis.

	Demographic potential growth rate - r	Long-term annual population growth rate - r_{long}	Effects of reproductive skew and poaching
Age-specific vital rates	✓	✓	✓
Environmental stochasticity		✓	✓

Demographic stochasticity		✓	✓
Density dependence		✓	✓
Reproductive skew		✓	✓ (with and without)
Poaching pressure			✓

Table 1. A summary of the variables included in each analysis

3. Results

3.1 Age-specific demographic parameters and differences between reserves

The profile of age-specific birth and mortality rates is typical for long-lived mammals, with relatively high mortality rates for very young and very old individuals (Supplementary Information S5). There are four peaks in birth rates, around 8, 15, 23 and 32 years of age, and rapid reproductive senescence after 32. A peak in reproduction just before senescence is unexpected but may be an artefact of lower numbers of older individuals in the dataset. All three reserves exhibited similar age-specific profiles of birth and death rates, but Ol Jogi on average showed higher average mortality rates and lower birth rates than the other two, where birth rates easily exceeded mortality rates (Supplementary Information S5).

A comparison of r and r_{long} values highlights differences between the reserves and the effect of stochasticity and reproductive skew on the projected growth rates (Table 2). Including these processes decreases the growth rates from the demographic potential of all three populations, and for Ol Jogi a positive r was accompanied by a negative r_{long} over 100 years. Lewa displayed the highest demographic potential growth rate, Ol Pejeta the highest long-term growth rate and Ol Jogi had much lower values for both..

Reserve	r	r_{long}
Lewa	0.0555	0.0055 ± 0.0034
Ol Jogi	0.0196	-0.0014 ± 0.0058
Ol Pejeta	0.0547	0.0066 ± 0.0028

Table 2. Intrinsic rates of increase (r) (calculated from Leslie matrices) and estimated population growth rates calculated from the simulations (r_{long}) with 95% confidence intervals.

Figure 1 – Baseline projections with reproductive skew. Projected changes in the sizes of the populations 100 years into the future. The solid lines show the mean of all 500 simulations and dotted lines show 95% confidence intervals

The range and mean of population sizes for projections with stochasticity and reproductive skew across 500 simulations varied across reserves (Figure 1). On average, the populations on Lewa and Ol Pejeta are predicted to continue to increase over 100 years, exceeding the predicted ECCs of 50 and 70 respectively. On Ol Jogi, the population is predicted to decline slowly.

3.3 Reproductive skew and population dynamics

There was substantial skew in reproductive success across individuals and there were different patterns of skew between the reserves (Supplementary Information S6). The overall reproductive skew distribution shows that black rhino females in these populations do vary in their reproductive success and suggests not all the female reproductive potential is being realised. This was most

strongly evident on Ol Pejeta and then Ol Jogi, whereas Lewa's distribution was more symmetric around the mean.

Generally, including reproductive skew in the model has a significant effect on the projected change in the population (Figure 2). With reproductive skew included the population size on Ol Jogi is lower than the 2019 size on average after 100 years. There was a lesser effect on Lewa and Ol Pejeta (Supplementary Information S7). The inclusion of female reproductive skew in population models is an important consideration, as it can increase the extinction probability for small populations, or those with low intrinsic growth rates, even without offtake.

Figure 2 – A comparison of the population projections for Ol Jogi a) without and b) with female reproductive skew. Graphs for Lewa and Ol Pejeta can be found in Supplementary Information S7

Offtake had expected negative impacts on population persistence, with >50% probability of extinction when adult offtake was greater than 4% (Ol Jogi), 11% (Lewa) or 12% (Ol Pejeta). Crucially, the inclusion of reproductive skew significantly increased the probability of extinction (Figure 3). Even with no offtake, 1.2% of the Ol Jogi simulations went extinct over 100 years when reproductive skew was included. At 5% offtake, the probability increases from 13.6% to 77.8%. The larger populations were also affected by reproductive skew. At 10% offtake, the extinction probabilities on Ol Pejeta and Lewa increased from 0.4% and 2.2% to 10.4% and 44.4% respectively when reproductive skew is included in the models.

Figure 3 – The change in proportion of 500 simulations that reach a population size of 0 over 100 years with and without reproductive skew under different levels of offtake. 95% confidence intervals calculated using the adjusted Wald technique

4. Discussion

Reproductive skew is an important factor to consider in studies of population dynamics [13] but it is rarely incorporated into PVAs. We provide evidence that variation in female breeding success can increase extinction risk from poaching. Datasets that allow for the estimation of variation in breeding success are rare, but PVAs which do not include it may be underestimating extinction risk. This work also highlights important differences between three key Kenyan black rhino reserves.

The estimated maximum long term intrinsic growth rate of a black rhino population is 9%-11% pa [43]. To achieve this, and allow the use of “established populations as a ‘breeding bank’ ... to build up other populations and to expand into new secure areas with suitable habitats” [23], the underlying causes of variation must be identified.

The vulnerability of the Ol Jogi population to extinction when faced with stochasticity and female reproductive skew is likely to be a consequence of both its smaller population size [54] and lower intrinsic growth rate. The greater variability in the long-term annual population growth rate (r_{long}) of Ol Jogi is likely due to the long period that the projections spend oscillating close to a maximum size. We ran projections using Ol Jogi vital rates but the starting population structure and carrying capacity of Ol Pejeta and no simulations went extinct but the mean population size was significantly lower than the other two reserves, at around 54 after 100 years. As this study focused on real-world populations, we did not vary population size, growth rates, or reproductive skew to investigate how their impact on extinction risk varied with their magnitude. Future work with simulated populations could vary these factors independently to test their relative importance.

Determining the cause of reserve-level differences is crucial for the conservation of black rhinos in Kenya. One possibility is that population performance is affected by differences in historical reproductive skew. However, Ol Pejeta seems to have the greatest skew, and the highest proportion of nulliparous or very slow breeding females (Supplementary Information 6). Differences in habitat variability could cause population-level differences. However, black rhino populations are relatively resilient to drought, potentially due to their browsing diet [12,55]. Our estimations of environmental stochasticity are based on observed variations in annual vital rates, but it could be that our model over-emphasises the role of environmental variation when social context could be the cause. Assigning causality to temporal variations in vital rates is a key challenge for future research.

Whilst the study reserves contain broadly similar habitat, demographic differences could be caused by fine scale habitat differences. This includes disease [56,57], predation [58] and diet availability. Assessments estimate that Ol Jogi has a higher proportion of browse that is considered highly suitable for black rhinos, so its lower intrinsic growth rate is surprising [59]. Whilst there is evidence for black rhino dietary preferences [60,61], new methods including metabarcoding [62] could be

used to directly link dietary composition with fitness. Although *D. b. michaeli* has retained the most genetic diversity of the remaining subspecies of black rhino [63], lower heterozygosity has been linked to reduced male reproductive success [35]. Low diversity could be affecting populations in unknown ways, and there are suspicions that bilateral blindness in Kenyan black rhinos could have a genetic component.

Large mammal species can be considered to be meta-populations if they have discreet breeding subpopulations with different growth rates and demographic fates [64]. We provide evidence that these black rhino populations fulfil these criteria. The physical, social and economic infrastructure required to maintain rhino reserves makes it unfeasible to move all individuals to optimal habitat and maintaining populations in different habitat types reduces overall extinction risk of a metapopulation [65]. Biological management, including the translocation of high value females that takes account of genetics, could lower the risk of extinction on reserves like Ol Jogi. If future work can explain the differences in population growth rates between the three reserve, then conservation planning for black rhinos could take account of factors that increase death rates and decrease birth rates.

Male reproductive skew has been found to have a small impact on the extinction risk of mammal populations affected by poaching [66]. Although male reproductive skew does not directly impact population dynamics, it may have impacts on long term genetic variation [67]. It should be noted that lower effective population size caused by reproductive skew in black rhinos may be compensated for to some extent by higher heterozygosity of dominant males [35]. We assumed that male availability does not affect breeding rates, but it could be important at low densities. It is likely that low densities of males and difficulties finding mates would exacerbate female reproductive skew and its population level effects, which would not invalidate our conclusions.

Female reproductive skew may be important in the short and long term even though it is often overlooked in non-cooperatively breeding species. We show that reproductive skew can affect projected population growth, particularly in small populations or those with low intrinsic growth rates, and provide empirical evidence for the theoretical proposition that structured variation in fecundity probabilities can increase extinction risk [68]. Even if PVAs are overly pessimistic [69], understanding the additive impact of skew on stochasticity is fundamentally important. It has far-reaching implications for conservation and the estimation of extinction risk, particularly for species that are affected by offtake.

As far as we are aware, the combined impact of variation in female breeding success and poaching has not been investigated in a large-bodied vertebrate species. It has been found that poaching

female adult giant pandas (*Ailuropoda melanoleuca*), rather than adult males or young individuals, leads to lower population sizes and a higher chance of extinction [70]. Here we have found that the level of female reproductive skew present in our study populations significantly increases the extinction risk of populations of large mammals that are affected by poaching. This may be because a large proportion of total reproductive potential is invested in relatively few individuals. Without poaching, these very fit individuals may prevent extinction [14] and would expect 'frail' individuals to be disproportionately lost, leading to higher growth rates and more resilient cohorts over time [8,9]. Poaching, however, is not selective and is likely to remove fit individuals. This will have a big impact on the growth of the population and destabilise cohorts. A crucial research priority is to find the level of female reproductive skew which starts to impact population dynamics. This will vary depending on environmental and social factors, but an estimated threshold would be very useful for management.

Datasets that allow for the estimation of variation in breeding success are rare. However, we suggest that the lack of research into the combination of poaching and reproductive skew on population dynamics may cause the underestimation of extinction risk for populations affected by poaching.

With respect to the conservation of black rhinos, the importance of female reproductive skew has two implications. Firstly, although many factors influence extinction risk, even low levels of poaching have the potential to damage the long-term viability of black rhino populations [71]. Tackling poaching is already a priority for Kenya, which aims to keep levels less than 1% per annum [23]. Secondly, management should be used to reduce female reproductive skew and encourage as many females as possible to breed. The methods will depend on the causes of reproductive skew, whether fitter females are able to monopolise the best resources, resist disease and predation, take dominant roles in social hierarchies or resist harassment from others. Future research should focus on factors that decrease the probability of females breeding and raising young, especially drivers of nulliparity or persistently poor breeding performance. For example, if reproductive skew is largely caused by differences in diet quality on the home-range scale, then habitat management could be used. This may be difficult in many places, especially if the abundance of preferred food plants is limited by rainfall [46], and supplementary feeding is often controversial. Placing new reserves in optimal habitat may be more feasible. On the other hand, if male harassment is preventing breeding in male-biased populations, then strategic metapopulation management including the removal of males could even out the sex ratio. Whilst translocations pose risks, biological management of black rhinos is routine. The difficulty lies in where to put excess males if many reserves struggle with a male-biased sex ratio.

Understanding the causes of reproductive skew will allow for it to change as a function of population features in population models. Poaching may actually set up feedback loops that worsen reproductive skew and lead to faster population declines. Poaching had indirect effects on demography in Kruger National Park due to reduced mate-finding as an Allee effect, disturbed social dynamics or increased calf predation [52]. Although our study reserves are much smaller than Kruger, female black rhinos change their spatial organisation very slowly after the death of a neighbouring individual [30], so poaching may decrease encounters with males and extended re-establishment of male dominance may make females reluctant to mate [52].

The demographic importance of female reproductive skew poses a difficult problem for the conservation of other species. Assessing whether a species exhibits reproductive skew requires long-term data collection on an individual level, which is difficult and expensive. Designing conservation programmes to mitigate the impact of female reproductive skew is even more challenging. Apart from tackling poaching and providing optimal habitat conditions, conservation specifically focused on alleviating reproductive skew must be done on an individual basis, including encouraging reproduction in females with low success or strategic biological management of metapopulations. The individual-level data and monitoring available for black rhino provide a way forward for assessing and mitigating the effect of skew but represent important knowledge gaps in the conservation of other species.

Ethics

This project was approved by the University of Manchester's Committee for the ethical review of category D research (Ref: 0030). Research was conducted in affiliation with the Kenya Wildlife Service and licensed by the Republic of Kenya's National Commission for Science & Innovation (Permit numbers: NACOSTI/P/17/87006/16178 and NACOSTI/P/19/1947).

References

1. IPBES. 2019 Global assessment report on biodiversity and ecosystem services of the Intergovernmental Science-Policy Platform on Biodiversity and Ecosystem Services.
2. WWF. 2020 Living Planet Report 2020 - Bending the curve of biodiversity loss.
3. Caughley G. 1994 Directions in Conservation Biology. *J. Anim. Ecol.* **63**, 215. (doi:<https://doi.org/10.2307/5542>)
4. Lande R, Engen S, Saether B. 2003 *Stochastic population dynamics in ecology and conservation*. Oxford: Oxford University Press.

5. O'Grady J, Brook B, Reed D, Ballou J, Tonkyn D, Frankham R. 2006 Realistic levels of inbreeding depression strongly affect extinction risk in wild populations. *Biol. Conserv.* **133**, 42–51. (doi:<https://doi.org/10.1016/j.biocon.2006.05.016>)
6. Berec L, Angulo E, Courchamp F. 2007 Multiple Allee effects and population management. *Trends Ecol. Evol.* **22**, 185–191. (doi:[10.1016/j.tree.2006.12.002](https://doi.org/10.1016/j.tree.2006.12.002))
7. Melbourne B, Hastings A. 2008 Extinction risk depends strongly on factors contributing to stochasticity. *Nature* **454**, 100–103. (doi:<https://doi.org/10.1038/nature06922>)
8. Kendall BE, Fox GA, Fujiwara M, Nogeire TM. 2011 Demographic heterogeneity, cohort selection, and population growth. *Ecology* **92**, 1985–1993. (doi:[10.1890/11-0079.1](https://doi.org/10.1890/11-0079.1))
9. Stover JP, Kendall BE, Fox GA. 2012 Demographic heterogeneity impacts density-dependent population dynamics. *Theor. Ecol.* **5**, 297–309. (doi:[10.1007/s12080-011-0129-x](https://doi.org/10.1007/s12080-011-0129-x))
10. Emlen ST. 1982 The Evolution of Helping. I. An Ecological Constraints Model. *Am. Nat.* **119**, 29–39. (doi:[10.1086/283888](https://doi.org/10.1086/283888))
11. Emlen ST. 1982 The Evolution of Helping. II. The Role of Behavioral Conflict. *Am. Nat.* **119**, 40–53. (doi:[10.1086/283889](https://doi.org/10.1086/283889))
12. Ferreira SM, Roex N le, Greaver C. 2019 Species-specific drought impacts on black and white rhinoceroses. *PLOS ONE* **14**, e0209678. (doi:[10.1371/journal.pone.0209678](https://doi.org/10.1371/journal.pone.0209678))
13. Kendall B, Fox G. 2003 Unstructured individual variation and demographic stochasticity. *Conserv. Biol.* **17**, 1170–1172.
14. Conner MM, White GC. 1999 Effects of Individual Heterogeneity in Estimating the Persistence of Small Populations. *Nat. Resour. Model.* **12**, 109–127. (doi:<https://doi.org/10.1111/j.1939-7445.1999.tb00005.x>)
15. Kendall BE, Fox GA. 2002 Variation among Individuals and Reduced Demographic Stochasticity. *Conserv. Biol.* **16**, 109–116. (doi:<https://doi.org/10.1046/j.1523-1739.2002.00036.x>)
16. Keller L, Reeve HK. 1994 Partitioning of reproduction in animal societies. *Trends Ecol. Evol.* **9**, 98–102. (doi:[https://doi.org/10.1016/0169-5347\(94\)90204-6](https://doi.org/10.1016/0169-5347(94)90204-6))
17. Ellis L. 1995 Dominance and reproductive success among nonhuman animals: A cross-species comparison. *Ethol. Sociobiol.* **16**, 257–333. (doi:[https://doi.org/10.1016/0162-3095\(95\)00050-U](https://doi.org/10.1016/0162-3095(95)00050-U))
18. Stockley P, Bro-Jørgensen J. 2011 Female competition and its evolutionary consequences in mammals. *Biol. Rev.* **86**, 341–366. (doi:<https://doi.org/10.1111/j.1469-185X.2010.00149.x>)
19. Clutton-Brock TH, Guinness FE, Albon SD. 1982 *Red deer: behavior and ecology of two sexes*. Chicago: University of Chicago press.
20. Archie EA, Chiyo PI. 2012 Elephant behaviour and conservation: social relationships, the effects of poaching, and genetic tools for management. *Mol. Ecol.* **21**, 765–778. (doi:<https://doi.org/10.1111/j.1365-294X.2011.05237.x>)
21. Milner-Gulland EJ, Bukreeva OM, Coulson T, Lushchekina AA, Kholodova MV, Bekenov AB, Grachev IA. 2003 Reproductive collapse in saiga antelope harems. *Nature* **422**, 135–135. (doi:<https://doi.org/10.1038/422135a>)

22. Knight M. 2018 African Rhino Specialist Group report/ Rapport du Groupe de Spécialistes du Rhinocéros d'Afrique. *Pachyderm* **59**, 14–26.
23. KWS. 2017 Kenyan Black Rhino Action Plan 2017-2021 Sixth Edition.
24. Edwards KL, Walker SL, Dunham AE, Pilgrim M, Okita-Ouma B, Shultz S. 2015 Low birth rates and reproductive skew limit the viability of Europe's captive eastern black rhinoceros, *Diceros bicornis michaeli*. *Biodivers. Conserv.* **24**, 2831–2852. (doi:<https://doi.org/10.1007/s10531-015-0976-7>)
25. Patton F, Campbell P, Parfet E. 2008 Biological management of the high density black rhino population in Solio Game Reserve, central Kenya. *Pachyderm* **44**, 72–79.
26. Law PR, Fike B, Lent PC. 2013 Mortality and female fecundity in an expanding black rhinoceros (*Diceros bicornis minor*) population. *Eur. J. Wildl. Res.* **59**, 477–485. (doi:[10.1007/s10344-013-0694-y](https://doi.org/10.1007/s10344-013-0694-y))
27. Göttert T, Schöne J, Zinner D, Hodges JK, Böer M. 2010 Habitat use and spatial organisation of relocated black rhinos in Namibia. **74**, 35–42. (doi:[10.1515/mamm.2010.012](https://doi.org/10.1515/mamm.2010.012))
28. Emslie R, Amin R, Kock R. 2009 Guidelines for the in situ re-introduction and translocation of African and Asian rhinoceros.
29. Gedir JV, Law PR, Preez P du, Linklater WL. 2018 Effects of age and sex ratios on offspring recruitment rates in translocated black rhinoceros. *Conserv. Biol.* **32**, 628–637. (doi:[10.1111/cobi.13029](https://doi.org/10.1111/cobi.13029))
30. Linklater WL, Hutcheson IR. 2010 Black Rhinoceros are Slow to Colonize a Harvested Neighbour's Range. *South Afr. J. Wildl. Res.* **40**, 58–63. (doi:[10.3957/056.040.0107](https://doi.org/10.3957/056.040.0107))
31. Boyce MS. 1992 Population viability analysis. *Annu. Rev. Ecol. Syst.* **23**, 481–497.
32. White GC. 2000 Population viability analysis: data requirements and essential analyses. In *Research techniques in animal ecology: controversies and consequences*. (eds L Boitani, T Fuller), pp. 288–331. New York, NY: Columbia University Press.
33. Frick WF *et al.* 2017 Fatalities at wind turbines may threaten population viability of a migratory bat. *Biol. Conserv.* **209**, 172–177. (doi:<https://doi.org/10.1016/j.biocon.2017.02.023>)
34. Chaudhary V, Oli MK. 2020 A critical appraisal of population viability analysis. *Conserv. Biol.* **34**, 26–40. (doi:<https://doi.org/10.1111/cobi.13414>)
35. Cain B, Wandera A, Shawcross S, Edwin-Harris W, Stevens-Wood B, Kemp S, Okita-Ouman B, Watts P. 2014 Sex-Biased Inbreeding Effects on Reproductive Success and Home Range Size of the Critically Endangered Black Rhinoceros. *Conserv. Biol.* **28**, 594–603. (doi:<https://doi.org/10.1111/cobi.12175>)
36. Garnier J, Bruford M, Goossens B. 2001 Mating system and reproductive skew in the black rhinoceros. *Mol. Ecol.* **10**, 2031–2041. (doi:<https://doi.org/10.1046/j.0962-1083.2001.01338.x>)
37. Caswell H. 2000 *Matrix population models*. Sinauer Sunderland, MA, USA.
38. Jackson J, Childs DZ, Mar KU, Htut W, Lummaa V. 2019 Long-term trends in wild-capture and population dynamics point to an uncertain future for captive elephants. *Proc. R. Soc. B Biol. Sci.* **286**, 20182810. (doi:<https://doi.org/10.1098/rspb.2018.2810>)

39. R Core Team. 2017 R: A language and environment for statistical computing.
40. Wood SN. 2011 Fast stable restricted maximum likelihood and marginal likelihood estimation of semiparametric generalized linear models. *J. R. Stat. Soc. Ser. B Stat. Methodol.* **73**, 3–36. (doi:<https://doi.org/10.1111/j.1467-9868.2010.00749.x>)
41. Hartig F. 2018 *DHARMA: residual diagnostics for hierarchical (multi-level/ mixed) regression models*. See <https://CRAN.R-project.org/package=DHARMA>.
42. Law PR, Fike B, Lent PC. 2015 Dynamics of an expanding black rhinoceros (*Diceros bicornis minor*) population. *Eur. J. Wildl. Res.* **61**, 601–609. (doi:<https://doi.org/10.1007/s10344-015-0935-3>)
43. Okita-Ouma B, Amin R, van Langevelde F, Leader-Williams N. 2010 Density dependence and population dynamics of black rhinos (*Diceros bicornis michaeli*) in Kenya’s rhino sanctuaries. *Afr. J. Ecol.* **48**, 791–799. (doi:<https://doi.org/10.1111/j.1365-2028.2009.01179.x>)
44. Fritz H, Duncan P. 1994 On the carrying capacity for large ungulates of African savanna ecosystems. *Proc. R. Soc. Lond. B Biol. Sci.* **256**, 77–82. (doi:<https://doi.org/10.1098/rspb.1994.0052>)
45. Birkett A. 2002 The impact of giraffe, rhino and elephant on the habitat of a black rhino sanctuary in Kenya. *Afr. J. Ecol.* **40**, 276–282. (doi:<https://doi.org/10.1046/j.1365-2028.2002.00373.x>)
46. Birkett A, Stevens-Wood B. 2005 Effect of low rainfall and browsing by large herbivores on an enclosed savannah habitat in Kenya. *Afr. J. Ecol.* **43**, 123–130. (doi:<https://doi.org/10.1111/j.1365-2028.2005.00555.x>)
47. McLeod SR. 1997 Is the Concept of Carrying Capacity Useful in Variable Environments? *Oikos* **79**, 529–542. (doi:<https://doi.org/10.2307/3546897>)
48. Ferreira SM, Greaver CC, Knight MH. 2011 Assessing the Population Performance of the Black Rhinoceros in Kruger National Park. *South Afr. J. Wildl. Res.* **41**, 192–204. (doi:[10.3957/056.041.0206](https://doi.org/10.3957/056.041.0206))
49. McCullough DR. 1992 Concepts of Large Herbivore Population Dynamics. In *Wildlife 2001: Populations* (eds DR McCullough, RH Barrett), pp. 967–984. Dordrecht: Springer Netherlands. (doi:https://doi.org/10.1007/978-94-011-2868-1_74)
50. Sibly RM, Barker D, Denham MC, Hone J, Pagel M. 2005 On the Regulation of Populations of Mammals, Birds, Fish, and Insects. *Science* **309**, 607–610. (doi:[10.1126/science.1110760](https://doi.org/10.1126/science.1110760))
51. Owen-Smith R. 1992 *Megaherbivores: the influence of very large body size on ecology*. Cambridge: Cambridge University Press.
52. Roex N le, Ferreira SM. 2020 Age structure changes indicate direct and indirect population impacts in illegally harvested black rhino. *PLOS ONE* **15**, e0236790. (doi:[10.1371/journal.pone.0236790](https://doi.org/10.1371/journal.pone.0236790))
53. Grimm V *et al.* 2006 A standard protocol for describing individual-based and agent-based models. *Ecol. Model.* **198**, 115–126. (doi:[10.1016/j.ecolmodel.2006.04.023](https://doi.org/10.1016/j.ecolmodel.2006.04.023))
54. Lacy RC. 2000 Considering Threats to the Viability of Small Populations Using Individual-Based Models. *Ecol. Bull.* , 39–51.

55. Roex N, Ferreira SM. 2021 Rhino birth recovery and resilience to drought impact. *Afr. J. Ecol.* **59**, 544–547. (doi:10.1111/aje.12854)
56. Fischer-Tenhagen C, Hamblin C, Quandt S, Frölich K. 2000 Serosurvey for selected infectious disease agents in free-ranging black and white rhinoceros in Africa. *J. Wildl. Dis.* **36**, 316–323. (doi:https://doi.org/10.7589/0090-3558-36.2.316)
57. Ndeereh D, Ouma BO, Gaymer J, Mutinda M, Gakuya F. 2012 Unusual mortalities of the eastern black rhinoceros (*Diceros bicornis michaeli*) due to clostridial enterotoxaemia in Ol Jogi Pyramid Sanctuary, Kenya. *Pachyderm* **51**, 45–51.
58. Patton F. 2009 Lion predation on the African Black Rhinoceros and its potential effect on management. *Endanger. Species Update* **26**, 43–50.
59. Adcock K, Amin R, Khayale C. 2007 Habitat characteristics and carrying capacity relationships of 9 Kenyan black rhino areas.
60. Muya SM, Oguge NO. 2000 Effects of browse availability and quality on black rhino (*Diceros bicornis michaeli* Groves 1967) diet in Nairobi National Park, Kenya. *Afr. J. Ecol.* **38**, 62–71. (doi:https://doi.org/10.1046/j.1365-2028.2000.00213.x)
61. Buk KG, Knight MH. 2010 Seasonal diet preferences of black rhinoceros in three arid South African National Parks. *Afr. J. Ecol.* **48**, 1064–1075. (doi:https://doi.org/10.1111/j.1365-2028.2010.01213.x)
62. Gill BA, Musili PM, Kurukura S, Hassan AA, Goheen JR, Kress WJ, Kuzmina M, Pringle RM, Kartzinel TR. 2019 Plant DNA-barcode library and community phylogeny for a semi-arid East African savanna. *Mol. Ecol. Resour.* **19**, 838–846. (doi:https://doi.org/10.1111/1755-0998.13001)
63. Harley E, Baumgarten I, Cunningham J, O’Ryan C. 2005 Genetic variation and population structure in remnant populations of black rhinoceros, *Diceros bicornis*, in Africa. *Mol. Ecol.* **14**, 2981–2990. (doi:https://doi.org/10.1111/j.1365-294X.2005.02660.x)
64. Elmhagen B, Angerbjörn A. 2001 The applicability of metapopulation theory to large mammals. *Oikos* **94**, 89–100. (doi:10.1034/j.1600-0706.2001.11316.x)
65. Reed D. 2004 Extinction risk in fragmented habitats. *Anim. Conserv. Forum* **7**, 181–191.
66. Horev A, Yosef R, Tryjanowski P, Ovadia O. 2012 Consequences of variation in male harem size to population persistence: Modeling poaching and extinction risk of Bengal tigers (*Panthera tigris*). *Biol. Conserv.* **147**, 22–31. (doi:https://doi.org/10.1016/j.biocon.2012.01.012)
67. Trask AE, Bignal EM, McCracken DI, Pieltney SB, Reid JM. 2017 Estimating demographic contributions to effective population size in an age-structured wild population experiencing environmental and demographic stochasticity. *J. Anim. Ecol.* **86**, 1082–1093. (doi:https://doi.org/10.1111/1365-2656.12703)
68. Robert A, Sarrazin F, Couvet D. 2003 Variation among Individuals, Demographic Stochasticity, and Extinction: Response to Kendall and Fox. *Conserv. Biol.* **17**, 1166–1169.
69. Brook BW. 2000 Pessimistic and Optimistic Bias in Population Viability Analysis. *Conserv. Biol.* **14**, 564–566.

70. Yiming L, Zhongwei G, Qisen Y, Yushan W, Niemelä J. 2003 The implications of poaching for giant panda conservation. *Biol. Conserv.* **111**, 125–136. (doi:[https://doi.org/10.1016/S0006-3207\(02\)00255-0](https://doi.org/10.1016/S0006-3207(02)00255-0))
71. Ferreira SM, Greaver C, Knight GA, Knight MH, Smit IPJ, Pienaar D. 2015 Disruption of Rhino Demography by Poachers May Lead to Population Declines in Kruger National Park, South Africa. *PLOS ONE* **10**, e0127783. (doi:<https://doi.org/10.1371/journal.pone.0127783>)

Supplementary Information S3 - Calculation of long-term population growth rate

Demographic potential

We created Leslie matrices for each reserve from the estimated age-specific vital rates and then using the *popbio* package [1] we calculated the asymptotic population growth rate, λ , for each population. Considering a Leslie matrix \mathbf{A} , the individual elements of the matrix a_{ij} give the transitions of individuals at age j to age i during a single year. λ is the dominant eigenvalue of \mathbf{A} , and the proportional rate of increase. The intrinsic rate of increase of the population r , which we term the demographic potential growth rate [2], is given by:

$$r = \ln \lambda$$

Long-term realised population growth rate

We calculated the proportional change, or ‘realised annual population growth rate’ between year t and year $t+1$ individually for each of the 500 simulations. We then plotted the mean and 95% confidence intervals of this value for each reserve over the 100 years of the population projections. We calculated the long-term realised population growth rate as the average from year 25 to year 100 and the average confidence intervals over the same time period, as transient dynamics in the early time steps of the simulations would have inflated population growth rates and at year 25 the growth rates entered a stochastic equilibrium (Figure S3.1). Here x is the long-term realised

population growth rate, and if $x > 1$ then the population will increase over time. The estimated long-term intrinsic rate of increase r_{long} , is given by:

$$r_{long} = \ln x$$

Figure S3.1 Average growth rates of the projections of each population over 500 simulations

a)

b)

c)

Figure S3.2 Average growth rates of the projections of over 500 simulations with 95% confidence intervals for a) Lewa, b) Ol Pejeta and c) Ol Jogi

References

1. Stubben CJ, Milligan BG. 2007 Estimating and Analyzing Demographic Models Using the popbio Package in R. *Journal of Statistical Software* **22**.
2. Caswell H. 2000 *Matrix population models*. Sinauer Sunderland, MA, USA.

Appendix D

You have done a good job at introducing the changes suggested by me and the referee. Overall, this version represent a great improvement over the revised one.

We would like to thank the editor and the reviewers for their comments and suggestions, we also believe that this work is much improved from previous iterations.

Nevertheless, there are a couple of minor comments from my part, including:

-lines 172-173: is there any reference that could be cited here?

We could not find an appropriate published reference for this. But it has been made clear that this was a condition of our research permits, and it is the policy of the IUCN African Rhino Specialist Group, lines 173-174. We did look for a AfRSG reference for this but it is only mentioned in passing on resources such as the Red List, so we thought that a mention of this policy in the text was more appropriate.

-lines 412-413: Here you could mention what the extinction probabilities are for each population under a 1% level of poaching.

This has been included in lines 413-415.

Some words have been taken out throughout to make space for the additions.

Female reproductive skew exacerbates the extinction risk from poaching in the eastern black rhino

Nick Harvey Sky^{ab}, John Jackson^c, Geoffrey Chege^d, Jamie Gaymer^e, David Kimiti^f, Samuel Mutisya^g,
Simon Nakito^g, Susanne Shultz^a

^a Department of Earth and Environmental Sciences, University of Manchester, Manchester, M13 9PL,
UK

^b North of England Zoological Society, Chester Zoo, Caughall Road, Chester, CH2 1LH, UK

^c Department of Zoology, University of Oxford, OX1 3SZ, UK

^d Lewa Wildlife Conservancy, P.O. Box, Private Bag, Isiolo-60300, Kenya

^e Ol Jogi Ltd., PO Box 259–10400, Nanyuki, Kenya

^f Grevy's Zebra Trust, PO Box 15351 – 00509, Nairobi, Kenya

§ Ol Pejeta Conservancy, PO Box 167, Nanyuki, Kenya

Corresponding author – Nick Harvey Sky nick.c.harvey@gmail.com

Acknowledgements

We express sincere thanks to the security, monitoring and research teams on all three reserves who meticulously collected the demographic data, especially I. Lemaiyan who was integral to this on Lewa. We would also like to thank F. Omengo and KWS for providing permission to conduct this study and to M. Kamau, A. Kibungei and B. Gituku for comments on an early version of the manuscript.

Funding

N.H.S. is funded by the NERC Manchester-Liverpool DTP, NE/L002469/1, and Chester Zoo as part of their Conservation Scholars programme

Abstract

Variation in individual demographic rates can have large consequences for populations. Female reproductive skew is an example of structured demographic heterogeneity where females have intrinsic qualities that make them more or less likely to breed. The consequences of reproductive skew for population dynamics are poorly understood in non-cooperatively breeding mammals, especially when coupled with other drivers such as poaching. We address this knowledge gap with population viability analyses using an age-specific, female-only, individual-based, stochastic population model built with long-term data for three Kenyan populations of the Critically Endangered eastern black rhino (*Diceros bicornis michaeli*). There was substantial reproductive skew, with a high proportion of females not breeding or doing so at very low rates. This had a large impact on the projected population growth rate for the smaller population on Ol Jogi. Moreover, including female reproductive skew exacerbates the effects of poaching, increasing the probability of extinction by ~70% under a simulated poaching pressure of 5% offtake per year. Tackling the effects of reproductive skew depends on whether it is mediated by habitat or social factors, with potential strategies including habitat and biological management respectively. Investigating and tackling reproductive skew in other species requires long-term, individual-level data collection.

Keywords

Population viability analysis, demographic variance, vital rates, reproductive skew, poaching, extinction risk

1. Introduction

With over 1 million species facing extinction [1] and a 68% average decline in monitored vertebrate populations between 1970-2016 [2] quantifying extinction risk and identifying its drivers in vulnerable populations is crucial for biodiversity conservation. Small populations are particularly vulnerable to extinction [3] because environmental and demographic stochasticity [4], inbreeding depression, loss of genetic diversity [5] and inverse density dependence (or Allee effects) [6], all become more important as population size decreases. Understanding the drivers of extinction and population dynamics in small populations is vital to the development of effective conservation strategies.

The causes of unequal realised vital rates (survival, growth and reproduction) across individuals that can disproportionately impact small populations can be grouped into different categories [7].

Demographic stochasticity is caused by random variation in probabilistic birth and death rates resulting in some individuals contributing more offspring or living longer by chance. Extrinsic factors contribute to environmental stochasticity; spatial or temporal variation in birth and death rates at the population level. In contrast to extrinsic factors and demographic stochasticity, which cause chance variation in realised reproductive rates between individuals, demographic heterogeneity is intrinsic variation in vital rates at the individual level, which may be due to genetic quality, maternal effects, access to resources or different exposure to stressors [8,9].

Reproductive skew, defined as unequal reproductive success between individuals of the same sex in a population [10,11], is one facet of demographic heterogeneity that may affect extinction risk differently to other variations in reproductive success. Reproductive skew is predicted to affect population growth independently of overall reproductive potential, population age structure and environmental changes [12]. Structured, as opposed to unstructured, variation in vital rates is not independent of the vital rates of other individuals or the identity of the individual [13]. It has been shown that structured demographic heterogeneity that is retained throughout individuals' lifetimes, or 'individual heterogeneity', can have significant effects on extinction risk in small populations [14]. Despite this, population models often incorporate demographic stochasticity, in which all individuals have equal vital rates but the chance of a demographic event is modified by sampling variance, but do not include demographic heterogeneity [15]. Disentangling age-specific reproduction from female reproductive skew and assessing their impacts on population viability will add important dynamics of individual heterogeneity into demographic studies.

For mammals there is often an assumption that, outside of cooperatively breeding species, there is little reproductive skew among females or it is not important [16,17]. Long-term datasets have shown that it is present in non-cooperatively breeding mammals [18,19] but few studies have investigated the effect on population viability.

Other drivers of population dynamics such as poaching may be magnify the effects of female reproductive skew. Poaching has been shown to increase male reproductive skew in African elephants (*Loxodonta africana*) [20] and catastrophic poaching of male saiga antelopes (*Saiga tatarica tatarica*) led to a crash in the number of pregnancies due to disturbed mating behaviour [21]. The combined effect of female reproductive skew and poaching on population viability has yet to be explored. If population persistence depends on the survival and reproduction of a small number of fecund individuals, then the poaching of those individuals may drastically increase extinction risk.

One Critically Endangered subspecies for which assessing extinction risk is vital is the eastern black rhino (*Diceros bicornis michaeli*) as poaching is still a major threat [22]. Kenya, the major range state for the subspecies, has set a target of 'net growth of at least 5% per annum maintained in at least six established populations' [23], which some reserves are not achieving [24]. To minimise poaching risk, Kenyan black rhinos are managed as an artificial meta-population [23]. Isolated populations present the opportunity to conduct proxy natural experiments for studying environmental and demographic drivers of population dynamics. Reproductive skew has been identified within both captive and free-living female black rhinos, including variation in the number of calves, age of first reproduction and inter-calving interval [24–26]. Intrinsic differences in quality may allow particular individuals to benefit from both environmental and social factors. Females are largely solitary, but regularly interact with other males and females which have adjacent or overlapping home ranges [27,28]. Dominant females may secure home ranges with better quality diets and breed more successfully. Intraspecific aggression, harassment from males or other females [29,30] and fighting [12], could cause stress and inhibit reproduction [18], with fitter females more likely to successfully harass others while resisting it themselves.

The aim of our study was to assess the local extinction risk of eastern black rhino populations in three Kenyan reserves under multiple drivers of population dynamics including poaching and reproductive skew. We constructed a population viability analysis (PVA), a method which is routinely used to assess the risk of extinction faced by a species or population over a particular time period [31,32] and the importance of particular threats [33]. We defined local extinction as a female population of zero, although populations would become functionally extinct before then. The utility

of PVAs to effectively quantify extinction risk and predict future population declines is reliant on parameterisation from high quality data and appropriate life-history assumptions [34]. Black rhinos in Kenya are intensively monitored as part of efforts to protect them from poaching and so there are excellent current and historical demographic data available.

We used a data-driven approach to estimate population viability using approximately 40 years of individual-based demographic data. Such long-term datasets are very rare, particularly for free-living populations of a Critically Endangered species. To explore extinction risk, we constructed an age-specific, female-only, individual-based, stochastic population model. An individual-based approach allowed us to include reproductive skew as an intrinsic reproductive score that was relative to the entire population, assigned to each individual at birth and stayed the same through their entire lives. As well as reproductive skew, the model incorporated density dependence, environmental stochasticity and demographic stochasticity. We then simulated population growth over the next 100 years for each reserve. Crucially, we used this as a case study to simulate population growth under different offtake scenarios to assess the effect that reproductive skew and different levels of poaching would have on population viability for a large mammal.

1. Methods

2.1 Study populations

We focused on three different Kenyan reserves. The 250km² Lewa Wildlife Conservancy in Meru County (0.20°N, 37.42°E) founded a 20km² rhino sanctuary in 1983 and converted completely to a conservancy in 1995. The 360km² Ol Pejeta Conservancy in Laikipia Country (0.02°N, 36.90°E) founded a rhino sanctuary in 1988. The 235km² Ol Jogi Conservancy in Laikipia County (0.32°N, 36.98°E) was established as a rhino sanctuary in 1980.

2.2 Data collection

Due to intensive monitoring to protect from poaching, there are high quality individual-based demographic data available. Black rhino calves usually stay with their mothers for around 2.5-3 years, making the assignment of maternity almost certain. Paternity is difficult to assign without genetic techniques [35] and we could not include males in this study. High skew in male breeding success [36] suggests a lot of extra breeding potential among males, so generally breeding is unlikely to be limited by male density. In mammalian population modelling, 'female dominance' is often assumed where there are always enough males to fertilise all females [37]. A female-only design was therefore appropriate.

The three reserves record the dates of births, deaths, imports (including the ages of imported individuals) and exports. The ages at which females died and gave birth has been accurately recorded since the foundation of each sanctuary. Data are available for the periods 1984-2019 for Lewa, 1980-2019 for Ol Jogi and 1990-2019 for Ol Pejeta.

We constructed a time-to-event demographic dataset for each population documenting whether each female died, bred, or was imported/exported in a given year since they were present in the population. For import and export events, individuals were brought in or removed from the population with no birth or death event. Individuals were only ever translocated once in their lifetimes. 10 females were imported to and 10 exported from Lewa, 19 to and two from Ol Jogi and 18 to and two from Ol Pejeta. Imported and non-imported individuals were not differentiated but we incorporated individual-level differences which capture some of the potential for variation between imported and non-imported individuals. The demographic outcomes were coded as binary response variables, where 1 indicated birth or survival in a given observation year. 36 calves died before their sex was recorded, eight on Lewa, 25 on Ol Pejeta and three on Ol Jogi. We randomly selected whether each of these was a male or a female and removed the simulated male calves from the model. There were then 99 females recorded from Ol Pejeta, 79 from Lewa, 55 from Ol Jogi and a total of 2252 year-age observations.

2.3 Estimation of age-specific vital rates

We constructed an age-specific model that incorporated vital rate changes across lifespan, an approach recently applied to Asian elephants (*Elephas maximus*) [38]. All analyses used R version 4.0.1 [39].

Mortality and birth events in the demographic records were used to quantify population vital rates. The proportion of females that died or gave birth to female calves at each specific age provided raw age-specific mortality and birth rates that did not include reproductive skew. The values used in the PVA were estimated from the raw data using generalised additive models (GAMs) implemented in the *mgcv* package [40]. The distributional assumptions were checked using the *DHARMA package* [41] (Supplementary Information S1).

2.4 Parameterisation of stochastic projection model

To assess the future viability of each population, we built female-only, stochastic individual-based projection models using the predicted age-specific vital rates and projected 100 years into the future. Projections started from the populations present at the end of 2019. We cannot present the age structures of these starting populations due to confidentiality of black rhino data which was a

condition of our research permits and is a policy of the IUCN SSC African Rhino Specialist Group. An individual-based modelling framework allowed us to incorporate demographic stochasticity, an important source of uncertainty, providing an advantage over deterministic models. Every year, individual death and reproduction were simulated using a Bernoulli distribution determined by the probabilities calculated using the GAMs for each reserve (a single trial for each living individual in each year) using the `rbinom()` function. Reproduction was dependent on survival and occurred after survival/death. We removed individuals over the age of 40 in each year of the simulations (probability of mortality for individuals aged 40 was given a value of 1), because few individuals survived over this age and there was large variation in life-history parameters. No individuals over the age of 40 reproduced in the dataset.

2.5 Estimate of environmental stochasticity

We estimated environmental stochasticity from observed variance in annual vital rates across the study period. We calculated the annual mortality rate for the whole population on each reserve, and the annual birth rate for reproductive ages (five to 34 years of age) from foundation to 2019. We then calculated the standard deviation of each of these annual vital rates, which represented the environmental stochasticity for each vital rate in each population. To incorporate these into the projections, every year of the simulation we sampled from two truncated normal distributions created using `rtruncnorm()`. These had a mean of zero, a standard deviation equivalent to that of the annual vital rates and were truncated at 0.5 and -0.5 to prevent unrealistic jumps in population size. The breeding and mortality probability of all individuals were modified separately every year by these simulated factors. After this we ensured that no individuals below the age of 5 bred, as it was assumed to be a pre-reproductive life stage.

2.6 Estimate of density dependence

Whilst there is evidence for declines in reproduction when black rhinos increase above habitat-specific densities [28], it is uncertain what density-dependent factors regulate their numbers [42,43]. In variable environments like African savanna carrying capacity is dependent on resource availability [44]. Increasing densities of rhinos can reduce diet availability through browsing pressure [45] but it primarily depends on rainfall [46]. The concept of a fixed ecological carrying capacity (ECC) is therefore not particularly meaningful in areas with variable rainfall density [47]. Intra- and intersexual competition may also be important in density-driven variations in vital rates [48]. Male harassment of females can decrease recruitment rates, and variation in fecundity can be driven by sex-ratio and density [30]. Social carrying capacity may be a more accurate description of a maximum density of rhinos than ECC. Regardless of the mechanism, black rhino populations cannot

grow in a limited area indefinitely, although pre-20th century maximum population densities are unknown. Therefore, there must be a way of including density dependence in a biologically relevant way.

Carrying capacity was estimated using the IUCN translocation guidelines which incorporate reserve size, average annual rainfall and habitat type [28]. We termed it ECC to be consistent with the IUCN even though it may be determined by both environmental and social factors. All three reserves fall into the category '0.2-0.4 rhino per km²'. We assumed that maximum density is 0.4/km², which gives ECC estimations of 100 for Lewa, 140 for Ol Pejeta and 90 for Ol Jogi. Assuming equal sex ratios, these were halved to give the predicted female ECCs.

We incorporated density dependence into the stochastic projection models using the hypotheses that density dependence only has a significant effect above 0.75(ECC) [28], and populations can increase above the estimated ECC [49]. We also mediated density dependence using environmental stochasticity. Theoretically, large, long-lived species exhibit convex relationships between population size and growth rate [50]. Below 0.75(ECC), environmental stochasticity was calculated as in section 2.5. If the population was a proportion x above 0.75(ECC), then the distributions from which environmental stochasticity was drawn were altered using $4(x - 0.75)$. The standard deviations were increased by adding $4(x - 0.75)$ to the standard deviation calculated in section 2.5. The mean of the sampling distributions for breeding and mortality probabilities were decreased or increased from zero by $4(x - 0.75)$ respectively. The distributions were truncated at -0.5 and 0.5 to prevent biologically unrealistic jumps in population size.

2.7 Reproductive skew

We conservatively estimated female reproductive skew by calculating the number of calves each female over the age of 9 had successfully raised to the age of one year. Generally, the earliest that black rhino females can calve is five years old [51], although some females have been recorded to calve between the ages of 4 and 5 [26]. The number of nulliparous females was dominated by younger individuals, and we considered that including all females over the age of five would inflate reproductive skew. We chose nine years as the average age of first calving is around seven [51] and nine is around the first peak of reproductive probability. The number of yearlings each female had produced was divided by their age above five, to give an annual rate of yearling production.

We calculated a reproductive score using both male and female calves as part of efforts to estimate it conservatively. With a 50:50 sex ratio, the skew would be the same as with just female calves. However, due to the relatively small sample size, using only female calves increased estimates of

reproductive skew. This distribution was only used to estimate the reproductive score of females relative to each other, and resulting values were scaled to conserve the average breeding probabilities calculated using the female-only vital rates. Males therefore do not feature in the model.

The distribution of reproductive success was created using `hist()`. At the start of the projection, or at birth, each female was assigned a relative reproductive score, that stayed with them throughout their lives, using the distribution of the rate of yearling production. A value was drawn from this distribution using `sample()`, assigning each individual an integer of one to ten, according to the probabilities from the distribution. All reproductive scores of new individuals were scaled around zero using `scale()`, to preserve the average breeding probability of each population. Reproductive values were divided by 100 so that the final highest modifications were an order of magnitude lower than the annual breeding probabilities of reproductive age females, which were between 0.1 and 0.2. Every year of the simulation, the probability that each individual female reproduced was altered by their relative reproductive score (Supplementary Information S2).

We assumed that the reproductive skew distribution stayed constant. If female reproductive skew is caused by social interactions such as harassment, or Allee effects [52], then density changes may affect it. However, as we could not attribute definite causes to reproductive variance, we could not confidently alter it as the social context changed.

A formal description of the model, following the protocol set out in Grimm et al. (2006) [53], can be found in Supplementary Information S2.

2.8 Analysis

2.8.1 Reproductive skew and poaching

We projected the change in population sizes with and without reproductive skew to test the impact that female reproductive skew has on each population under different poaching regimes. We used a Bernoulli distribution created with `rbinom()` to decide whether each adult over the age of five would be poached in a particular year. We refer to this as the percentage annual adult offtake. We increased offtake from 0% to 20%, which allowed us to compare how the probability of extirpation of each population over 100 years changed with and without reproductive skew under different simulated poaching regimes.

2.8.2 Long-term growth rates

In order to explore the effect of stochasticity, reproductive skew and density dependence on the projected growth rates, we first estimated the intrinsic rate of increase r of each population using Leslie matrices [37], which we term the demographic potential growth rate. We then explored the difference between demographic potential growth rates and long-run realised population growth rates accounting for environmental stochasticity, demographic stochasticity, density dependence and reproductive skew by calculating the ‘long-term annual population growth rate’ r_{long} using our stochastic projections. Full methods and mathematical justifications can be found in Supplementary Information S3.

Table 1 shows a summary of the variables included in each analysis. Supplementary Information S4 presents an elasticity analysis.

	Demographic potential growth rate - r	Long-term annual population growth rate - r_{long}	Effects of reproductive skew and poaching
Age-specific vital rates	✓	✓	✓
Environmental stochasticity		✓	✓
Demographic stochasticity		✓	✓
Density dependence		✓	✓
Reproductive skew		✓	✓ (with and without)
Poaching pressure			✓

Table 1. A summary of the variables included in each analysis

3. Results

3.1 Age-specific demographic parameters and differences between reserves

The profile of age-specific birth and mortality rates is typical for long-lived mammals, with relatively high mortality rates for very young and very old individuals (Supplementary Information S5). There are four peaks in birth rates, around 8, 15, 23 and 32 years of age, and rapid reproductive senescence after 32. The unexpected peak in reproduction just before senescence may be an artefact of lower numbers of older individuals in the dataset. All three reserves exhibited similar age-specific profiles of birth and death rates, but Ol Jogi on average showed higher average mortality rates and lower birth rates than the other two, where birth rates easily exceeded mortality rates (Supplementary Information S5).

A comparison of r and r_{long} values highlights differences between the reserves and the effect of stochasticity and reproductive skew on the projected growth rates (Table 2). Including these processes decreases the growth rates from the demographic potential of all three populations, and for Ol Jogi a positive r was accompanied by a negative r_{long} over 100 years. Lewa displayed the highest demographic potential growth rate, Ol Pejeta the highest long-term growth rate and Ol Jogi had much lower values for both.

Reserve	r	r_{long}
Lewa	0.0555	0.0055 ± 0.0034
Ol Jogi	0.0196	-0.0014 ± 0.0058
Ol Pejeta	0.0547	0.0066 ± 0.0028

Table 2. Intrinsic rates of increase (r) (calculated from Leslie matrices) and estimated population growth rates calculated from the simulations (r_{long}) with 95% confidence intervals.

Figure 1 – Baseline projections with reproductive skew. Projected changes in the sizes of the populations 100 years into the future. The solid lines show the mean of all 500 simulations and dotted lines show 95% confidence intervals

The ranges and means of population sizes of projections with stochasticity and reproductive skew across 500 simulations varied across reserves (Figure 1). On average, the populations on Lewa and Ol Pejeta are predicted to continue to increase over 100 years, exceeding the predicted ECCs of 50 and 70 respectively. On Ol Jogi, the population is predicted to decline slowly.

3.3 Reproductive skew and population dynamics

There was substantial skew in reproductive success across individuals and there were different patterns of skew between the reserves (Supplementary Information S6). The overall reproductive skew distribution shows that black rhino females in these populations do vary in their reproductive success and suggests not all the female reproductive potential is being realised. This was most strongly evident on Ol Pejeta and then Ol Jogi, whereas Lewa's distribution was more symmetric around the mean.

Generally, including reproductive skew in the model has a significant effect on the projected change in the population (Figure 2). With reproductive skew included the population size on Ol Jogi is lower than the 2019 size on average after 100 years. There was a lesser effect on Lewa and Ol Pejeta (Supplementary Information S7). Female reproductive skew can increase the extinction probability for small populations, or those with low intrinsic growth rates, even without offtake.

Figure 2 – A comparison of the population projections for OI Jogi a) without and b) with female reproductive skew. Graphs for Lewa and OI Pejeta can be found in Supplementary Information S7

Offtake had expected negative impacts on population persistence, with >50% probability of extinction when adult offtake was greater than 4% (OI Jogi), 11% (Lewa) or 12% (OI Pejeta). Crucially, the inclusion of reproductive skew significantly increased the probability of extinction (Figure 3). Even with no offtake, 1.2% of the OI Jogi simulations went extinct over 100 years when reproductive skew was included. At 5% offtake, the probability increases from 13.6% to 77.8%. The larger populations were also affected by reproductive skew. At 10% offtake, the extinction probabilities on

Ol Pejeta and Lewa increased from 0.4% and 2.2% to 10.4% and 44.4% respectively when reproductive skew is included in the models.

Figure 3 – The change in proportion of 500 simulations that reach a population size of 0 over 100 years with and without reproductive skew under different levels of offtake. 95% confidence intervals calculated using the adjusted Wald technique

4. Discussion

Reproductive skew is an important factor to consider in studies of population dynamics [13] but it is rarely incorporated into PVAs. We provide evidence that variation in female breeding success can increase extinction risk from poaching. Datasets that allow for the estimation of variation in breeding success are rare, but PVAs which do not include it may be underestimating extinction risk. This work also highlights important differences between three key Kenyan black rhino reserves.

The estimated maximum long term intrinsic growth rate of a black rhino population is 9%-11% pa [43]. To achieve this, and allow the use of “established populations as a ‘breeding bank’ ... to build up other populations and to expand into new secure areas with suitable habitats” [23], the underlying causes of variation must be identified.

The vulnerability of the Ol Jogi population to extinction when faced with stochasticity and female reproductive skew is likely to be a consequence of both its smaller population size [54] and lower intrinsic growth rate. The greater variability in the long-term annual population growth rate (r_{long}) of Ol Jogi is likely due to the long period that the projections spend oscillating close to a maximum size. We ran projections using Ol Jogi vital rates but the starting population structure and carrying capacity of Ol Pejeta and no simulations went extinct but the mean population size was significantly lower than the other two reserves, at around 54 after 100 years. As this study focused on real-world populations, we did not vary population size, growth rates, or reproductive skew to investigate how their impact on extinction risk varied with their magnitude. Future work with simulated populations could vary these factors independently to test their relative importance.

Determining the cause of reserve-level differences is crucial for the conservation of black rhinos in Kenya. One possibility is that population performance is affected by differences in historical reproductive skew. However, Ol Pejeta seems to have the greatest skew, and the highest proportion of nulliparous or very slow breeding females (Supplementary Information 6). Differences in habitat variability could cause population-level differences. However, black rhino populations are relatively resilient to drought, potentially due to their browsing diet [12,55]. Our estimations of environmental stochasticity are based on observed variations in annual vital rates, but it could be that our model over-emphasises the role of environmental variation when social context could be the cause. Assigning causality to temporal variations in vital rates is a key challenge for future research.

Whilst the study reserves contain broadly similar habitat, demographic differences could be caused by fine scale habitat differences. This includes disease [56,57], predation [58] and diet availability. Assessments estimate that Ol Jogi has a higher proportion of browse that is considered highly suitable for black rhinos, so its lower intrinsic growth rate is surprising [59]. Whilst there is evidence for black rhino dietary preferences [60,61], new methods including metabarcoding [62] could be used to directly link dietary composition with fitness. Although *D. b. michaeli* has retained the most genetic diversity of the remaining subspecies of black rhino [63], lower heterozygosity has been linked to reduced male reproductive success [35]. Low diversity could be affecting populations in unknown ways, and there are suspicions that bilateral blindness in Kenyan black rhinos could have a genetic component.

Large mammal species can be considered to be meta-populations if they have discreet breeding subpopulations with different growth rates and demographic fates [64]. We provide evidence that these black rhino populations fulfil these criteria. The physical, social and economic infrastructure required to maintain rhino reserves makes it unfeasible to move all individuals to optimal habitat

and maintaining populations in different habitat types reduces overall extinction risk of a metapopulation [65]. Biological management, including the translocation of high value females that takes account of genetics, could lower the risk of extinction on reserves like Ol Jogi. If future work can explain the differences in population growth rates between the three reserve, then conservation planning for black rhinos could take account of factors that increase death rates and decrease birth rates.

Male reproductive skew has been found to have a small impact on the extinction risk of mammal populations affected by poaching [66]. Although male reproductive skew does not directly impact population dynamics, it may have impacts on long term genetic variation [67]. It should be noted that lower effective population size caused by reproductive skew in black rhinos may be compensated for to some extent by higher heterozygosity of dominant males [35]. We assumed that male availability does not affect breeding rates, but it could be important at low densities. It is likely that low densities of males and difficulties finding mates would exacerbate female reproductive skew and its population level effects, which would not invalidate our conclusions.

Female reproductive skew may be important in the short and long term even though it is often overlooked in non-cooperatively breeding species. We show that reproductive skew can affect projected population growth, particularly in small populations or those with low intrinsic growth rates, and provide empirical evidence for the theoretical proposition that structured variation in fecundity probabilities can increase extinction risk [68]. Even if PVAs are overly pessimistic [69], understanding the additive impact of skew on stochasticity is fundamentally important. It has far-reaching implications for conservation and the estimation of extinction risk, particularly for species that are affected by offtake.

As far as we are aware, the combined impact of variation in female breeding success and poaching has not been investigated in a large-bodied vertebrate species. It has been found that poaching female adult giant pandas (*Ailuropoda melanoleuca*), rather than adult males or young individuals, leads to lower population sizes and a higher chance of extinction [70]. Here we have found that the level of female reproductive skew present in our study populations significantly increases the extinction risk of populations of large mammals that are affected by poaching. This may be because a large proportion of total reproductive potential is invested in relatively few individuals. Without poaching, these very fit individuals may prevent extinction [14] and we would expect 'frail' individuals to be disproportionately lost, leading to higher growth rates and more resilient cohorts over time [8,9]. Poaching, however, is not selective and is likely to remove fit individuals. This will have a big impact on the growth of the population and destabilise cohorts. A crucial research priority

is to find the level of female reproductive skew which starts to impact population dynamics. This will vary depending on environmental and social factors, but an estimated threshold would be very useful for management. Datasets that allow for the estimation of variation in breeding success are rare. However, we suggest that the lack of research into the combination of poaching and reproductive skew on population dynamics may cause the underestimation of extinction risk for populations affected by poaching.

With respect to the conservation of black rhinos, the importance of female reproductive skew has two implications. Firstly, although many factors influence extinction risk, even low levels of poaching have the potential to damage the long-term viability of black rhino populations [71]. Tackling poaching is already a priority for Kenya, which aims to keep levels less than 1% per annum [23]. At 1% offtake in our projections with reproductive skew, no simulations reached extinction over 100 years on Lewa or Ol Pejeta, but almost 5% did so on Ol Jogi. Secondly, management should be used to reduce female reproductive skew and encourage as many females as possible to breed. The methods will depend on the causes of reproductive skew, whether fitter females are able to monopolise the best resources, resist disease and predation, take dominant roles in social hierarchies or resist harassment from others. Future research should focus on factors that decrease the probability of females breeding and raising young, especially drivers of nulliparity or persistently poor breeding performance. For example, if reproductive skew is largely caused by differences in diet quality on the home-range scale, then habitat management could be used. This may be difficult in many places, especially if the abundance of preferred food plants is limited by rainfall [46], and supplementary feeding is often controversial. Placing new reserves in optimal habitat may be more feasible. On the other hand, if male harassment is preventing breeding in male-biased populations, then strategic metapopulation management including the removal of males could even out the sex ratio. Whilst translocations pose risks, biological management of black rhinos is routine. The difficulty lies in where to put excess males if many reserves struggle with a male-biased sex ratio.

Understanding the causes of reproductive skew will allow for it to change as a function of population features in population models. Poaching may actually set up feedback loops that worsen reproductive skew and lead to faster population declines. Poaching had indirect effects on demography in Kruger National Park due to reduced mate-finding as an Allee effect, disturbed social dynamics or increased calf predation [52]. Although our study reserves are much smaller than Kruger, female black rhinos change their spatial organisation very slowly after the death of a neighbouring individual [30], so poaching may decrease encounters with males and extended re-establishment of male dominance may make females reluctant to mate [52].

The demographic importance of female reproductive skew poses a difficult problem for the conservation of other species. Assessing whether a species exhibits reproductive skew requires long-term data collection on an individual level, which is difficult and expensive. Designing conservation programmes to mitigate the impact of female reproductive skew is even more challenging. Apart from tackling poaching and providing optimal habitat conditions, conservation specifically focused on alleviating reproductive skew must be done on an individual basis, including encouraging reproduction in females with low success or strategic biological management of metapopulations. The individual-level data and monitoring available for black rhino provide a way forward for assessing and mitigating the effect of skew but represent important knowledge gaps in the conservation of other species.

Ethics

This project was approved by the University of Manchester's Committee for the ethical review of category D research (Ref: 0030). Research was conducted in affiliation with the Kenya Wildlife Service and licensed by the Republic of Kenya's National Commission for Science & Innovation (Permit numbers: NACOSTI/P/17/87006/16178 and NACOSTI/P/19/1947).

References

1. IPBES. 2019 Global assessment report on biodiversity and ecosystem services of the Intergovernmental Science-Policy Platform on Biodiversity and Ecosystem Services.
2. WWF. 2020 Living Planet Report 2020 - Bending the curve of biodiversity loss.
3. Caughley G. 1994 Directions in Conservation Biology. *J. Anim. Ecol.* **63**, 215. (doi:<https://doi.org/10.2307/5542>)
4. Lande R, Engen S, Saether B. 2003 *Stochastic population dynamics in ecology and conservation*. Oxford: Oxford University Press.
5. O'Grady J, Brook B, Reed D, Ballou J, Tonkyn D, Frankham R. 2006 Realistic levels of inbreeding depression strongly affect extinction risk in wild populations. *Biol. Conserv.* **133**, 42–51. (doi:<https://doi.org/10.1016/j.biocon.2006.05.016>)
6. Berec L, Angulo E, Courchamp F. 2007 Multiple Allee effects and population management. *Trends Ecol. Evol.* **22**, 185–191. (doi:10.1016/j.tree.2006.12.002)
7. Melbourne B, Hastings A. 2008 Extinction risk depends strongly on factors contributing to stochasticity. *Nature* **454**, 100–103. (doi:<https://doi.org/10.1038/nature06922>)
8. Kendall BE, Fox GA, Fujiwara M, Nogueira TM. 2011 Demographic heterogeneity, cohort selection, and population growth. *Ecology* **92**, 1985–1993. (doi:10.1890/11-0079.1)
9. Stover JP, Kendall BE, Fox GA. 2012 Demographic heterogeneity impacts density-dependent population dynamics. *Theor. Ecol.* **5**, 297–309. (doi:10.1007/s12080-011-0129-x)
10. Emlen ST. 1982 The Evolution of Helping. I. An Ecological Constraints Model. *Am. Nat.* **119**, 29–39. (doi:10.1086/283888)
11. Emlen ST. 1982 The Evolution of Helping. II. The Role of Behavioral Conflict. *Am. Nat.* **119**, 40–53. (doi:10.1086/283889)
12. Ferreira SM, Roex N le, Greaver C. 2019 Species-specific drought impacts on black and white rhinoceroses. *PLOS ONE* **14**, e0209678. (doi:10.1371/journal.pone.0209678)

13. Kendall B, Fox G. 2003 Unstructured individual variation and demographic stochasticity. *Conserv. Biol.* **17**, 1170–1172.
14. Conner MM, White GC. 1999 Effects of Individual Heterogeneity in Estimating the Persistence of Small Populations. *Nat. Resour. Model.* **12**, 109–127. (doi:<https://doi.org/10.1111/j.1939-7445.1999.tb00005.x>)
15. Kendall BE, Fox GA. 2002 Variation among Individuals and Reduced Demographic Stochasticity. *Conserv. Biol.* **16**, 109–116. (doi:<https://doi.org/10.1046/j.1523-1739.2002.00036.x>)
16. Keller L, Reeve HK. 1994 Partitioning of reproduction in animal societies. *Trends Ecol. Evol.* **9**, 98–102. (doi:[https://doi.org/10.1016/0169-5347\(94\)90204-6](https://doi.org/10.1016/0169-5347(94)90204-6))
17. Ellis L. 1995 Dominance and reproductive success among nonhuman animals: A cross-species comparison. *Ethol. Sociobiol.* **16**, 257–333. (doi:[https://doi.org/10.1016/0162-3095\(95\)00050-U](https://doi.org/10.1016/0162-3095(95)00050-U))
18. Stockley P, Bro-Jørgensen J. 2011 Female competition and its evolutionary consequences in mammals. *Biol. Rev.* **86**, 341–366. (doi:<https://doi.org/10.1111/j.1469-185X.2010.00149.x>)
19. Clutton-Brock TH, Guinness FE, Albon SD. 1982 *Red deer: behavior and ecology of two sexes*. Chicago: University of Chicago press.
20. Archie EA, Chiyo PI. 2012 Elephant behaviour and conservation: social relationships, the effects of poaching, and genetic tools for management. *Mol. Ecol.* **21**, 765–778. (doi:<https://doi.org/10.1111/j.1365-294X.2011.05237.x>)
21. Milner-Gulland EJ, Bukreeva OM, Coulson T, Lushchekina AA, Kholodova MV, Bekenov AB, Grachev IA. 2003 Reproductive collapse in saiga antelope harems. *Nature* **422**, 135–135. (doi:<https://doi.org/10.1038/422135a>)
22. Knight M. 2018 African Rhino Specialist Group report/ Rapport du Groupe de Spécialistes du Rhinocéros d’Afrique. *Pachyderm* **59**, 14–26.
23. KWS. 2017 Kenyan Black Rhino Action Plan 2017-2021 Sixth Edition.
24. Edwards KL, Walker SL, Dunham AE, Pilgrim M, Okita-Ouma B, Shultz S. 2015 Low birth rates and reproductive skew limit the viability of Europe’s captive eastern black rhinoceros, *Diceros bicornis michaeli*. *Biodivers. Conserv.* **24**, 2831–2852. (doi:<https://doi.org/10.1007/s10531-015-0976-7>)
25. Patton F, Campbell P, Parfet E. 2008 Biological management of the high density black rhino population in Solio Game Reserve, central Kenya. *Pachyderm* **44**, 72–79.
26. Law PR, Fike B, Lent PC. 2013 Mortality and female fecundity in an expanding black rhinoceros (*Diceros bicornis minor*) population. *Eur. J. Wildl. Res.* **59**, 477–485. (doi:[10.1007/s10344-013-0694-y](https://doi.org/10.1007/s10344-013-0694-y))
27. Göttert T, Schöne J, Zinner D, Hodges JK, Böer M. 2010 Habitat use and spatial organisation of relocated black rhinos in Namibia. **74**, 35–42. (doi:[10.1515/mamm.2010.012](https://doi.org/10.1515/mamm.2010.012))
28. Emslie R, Amin R, Kock R. 2009 Guidelines for the in situ re-introduction and translocation of African and Asian rhinoceros.
29. Gedir JV, Law PR, Preez P du, Linklater WL. 2018 Effects of age and sex ratios on offspring recruitment rates in translocated black rhinoceros. *Conserv. Biol.* **32**, 628–637. (doi:[10.1111/cobi.13029](https://doi.org/10.1111/cobi.13029))
30. Linklater WL, Hutcheson IR. 2010 Black Rhinoceros are Slow to Colonize a Harvested Neighbour’s Range. *South Afr. J. Wildl. Res.* **40**, 58–63. (doi:[10.3957/056.040.0107](https://doi.org/10.3957/056.040.0107))
31. Boyce MS. 1992 Population viability analysis. *Annu. Rev. Ecol. Syst.* **23**, 481–497.
32. White GC. 2000 Population viability analysis: data requirements and essential analyses. In *Research techniques in animal ecology: controversies and consequences*. (eds L Boitani, T Fuller), pp. 288–331. New York, NY: Columbia University Press.
33. Frick WF *et al.* 2017 Fatalities at wind turbines may threaten population viability of a migratory bat. *Biol. Conserv.* **209**, 172–177. (doi:<https://doi.org/10.1016/j.biocon.2017.02.023>)

34. Chaudhary V, Oli MK. 2020 A critical appraisal of population viability analysis. *Conserv. Biol.* **34**, 26–40. (doi:<https://doi.org/10.1111/cobi.13414>)
35. Cain B, Wandera A, Shawcross S, Edwin-Harris W, Stevens-Wood B, Kemp S, Okita-Ouman B, Watts P. 2014 Sex-Biased Inbreeding Effects on Reproductive Success and Home Range Size of the Critically Endangered Black Rhinoceros. *Conserv. Biol.* **28**, 594–603. (doi:<https://doi.org/10.1111/cobi.12175>)
36. Garnier J, Bruford M, Goossens B. 2001 Mating system and reproductive skew in the black rhinoceros. *Mol. Ecol.* **10**, 2031–2041. (doi:<https://doi.org/10.1046/j.0962-1083.2001.01338.x>)
37. Caswell H. 2000 *Matrix population models*. Sinauer Sunderland, MA, USA.
38. Jackson J, Childs DZ, Mar KU, Htut W, Lummaa V. 2019 Long-term trends in wild-capture and population dynamics point to an uncertain future for captive elephants. *Proc. R. Soc. B Biol. Sci.* **286**, 20182810. (doi:<https://doi.org/10.1098/rspb.2018.2810>)
39. R Core Team. 2017 R: A language and environment for statistical computing.
40. Wood SN. 2011 Fast stable restricted maximum likelihood and marginal likelihood estimation of semiparametric generalized linear models. *J. R. Stat. Soc. Ser. B Stat. Methodol.* **73**, 3–36. (doi:<https://doi.org/10.1111/j.1467-9868.2010.00749.x>)
41. Hartig F. 2018 *DHARMA: residual diagnostics for hierarchical (multi-level/ mixed) regression models*. See <https://CRAN.R-project.org/package=DHARMA>.
42. Law PR, Fike B, Lent PC. 2015 Dynamics of an expanding black rhinoceros (*Diceros bicornis minor*) population. *Eur. J. Wildl. Res.* **61**, 601–609. (doi:<https://doi.org/10.1007/s10344-015-0935-3>)
43. Okita-Ouma B, Amin R, van Langevelde F, Leader-Williams N. 2010 Density dependence and population dynamics of black rhinos (*Diceros bicornis michaeli*) in Kenya's rhino sanctuaries. *Afr. J. Ecol.* **48**, 791–799. (doi:<https://doi.org/10.1111/j.1365-2028.2009.01179.x>)
44. Fritz H, Duncan P. 1994 On the carrying capacity for large ungulates of African savanna ecosystems. *Proc. R. Soc. Lond. B Biol. Sci.* **256**, 77–82. (doi:<https://doi.org/10.1098/rspb.1994.0052>)
45. Birkett A. 2002 The impact of giraffe, rhino and elephant on the habitat of a black rhino sanctuary in Kenya. *Afr. J. Ecol.* **40**, 276–282. (doi:<https://doi.org/10.1046/j.1365-2028.2002.00373.x>)
46. Birkett A, Stevens-Wood B. 2005 Effect of low rainfall and browsing by large herbivores on an enclosed savannah habitat in Kenya. *Afr. J. Ecol.* **43**, 123–130. (doi:<https://doi.org/10.1111/j.1365-2028.2005.00555.x>)
47. McLeod SR. 1997 Is the Concept of Carrying Capacity Useful in Variable Environments? *Oikos* **79**, 529–542. (doi:<https://doi.org/10.2307/3546897>)
48. Ferreira SM, Greaver CC, Knight MH. 2011 Assessing the Population Performance of the Black Rhinoceros in Kruger National Park. *South Afr. J. Wildl. Res.* **41**, 192–204. (doi:[10.3957/056.041.0206](https://doi.org/10.3957/056.041.0206))
49. McCullough DR. 1992 Concepts of Large Herbivore Population Dynamics. In *Wildlife 2001: Populations* (eds DR McCullough, RH Barrett), pp. 967–984. Dordrecht: Springer Netherlands. (doi:https://doi.org/10.1007/978-94-011-2868-1_74)
50. Sibly RM, Barker D, Denham MC, Hone J, Pagel M. 2005 On the Regulation of Populations of Mammals, Birds, Fish, and Insects. *Science* **309**, 607–610. (doi:[10.1126/science.1110760](https://doi.org/10.1126/science.1110760))
51. Owen-Smith R. 1992 *Megaherbivores: the influence of very large body size on ecology*. Cambridge: Cambridge University Press.
52. Roex N le, Ferreira SM. 2020 Age structure changes indicate direct and indirect population impacts in illegally harvested black rhino. *PLOS ONE* **15**, e0236790. (doi:[10.1371/journal.pone.0236790](https://doi.org/10.1371/journal.pone.0236790))
53. Grimm V *et al.* 2006 A standard protocol for describing individual-based and agent-based models. *Ecol. Model.* **198**, 115–126. (doi:[10.1016/j.ecolmodel.2006.04.023](https://doi.org/10.1016/j.ecolmodel.2006.04.023))

54. Lacy RC. 2000 Considering Threats to the Viability of Small Populations Using Individual-Based Models. *Ecol. Bull.* , 39–51.
55. Roex N, Ferreira SM. 2021 Rhino birth recovery and resilience to drought impact. *Afr. J. Ecol.* **59**, 544–547. (doi:10.1111/aje.12854)
56. Fischer-Tenhagen C, Hamblin C, Quandt S, Frölich K. 2000 Serosurvey for selected infectious disease agents in free-ranging black and white rhinoceros in africa. *J. Wildl. Dis.* **36**, 316–323. (doi:https://doi.org/10.7589/0090-3558-36.2.316)
57. Ndeereh D, Ouma BO, Gaymer J, Mutinda M, Gakuya F. 2012 Unusual mortalities of the eastern black rhinoceros (*Diceros bicornis michaeli*) due to clostridial enterotoxaemia in Ol Jogi Pyramid Sanctuary, Kenya. *Pachyderm* **51**, 45–51.
58. Patton F. 2009 Lion predation on the African Black Rhinoceros and its potential effect on management. *Endanger. Species Update* **26**, 43–50.
59. Adcock K, Amin R, Khayale C. 2007 Habitat characteristics and carrying capacity relationships of 9 Kenyan black rhino areas.
60. Muya SM, Oguge NO. 2000 Effects of browse availability and quality on black rhino (*Diceros bicornis michaeli* Groves 1967) diet in Nairobi National Park, Kenya. *Afr. J. Ecol.* **38**, 62–71. (doi:https://doi.org/10.1046/j.1365-2028.2000.00213.x)
61. Buk KG, Knight MH. 2010 Seasonal diet preferences of black rhinoceros in three arid South African National Parks. *Afr. J. Ecol.* **48**, 1064–1075. (doi:https://doi.org/10.1111/j.1365-2028.2010.01213.x)
62. Gill BA, Musili PM, Kurukura S, Hassan AA, Goheen JR, Kress WJ, Kuzmina M, Pringle RM, Kartzinel TR. 2019 Plant DNA-barcode library and community phylogeny for a semi-arid East African savanna. *Mol. Ecol. Resour.* **19**, 838–846. (doi:https://doi.org/10.1111/1755-0998.13001)
63. Harley E, Baumgarten I, Cunningham J, O’Ryan C. 2005 Genetic variation and population structure in remnant populations of black rhinoceros, *Diceros bicornis*, in Africa. *Mol. Ecol.* **14**, 2981–2990. (doi:https://doi.org/10.1111/j.1365-294X.2005.02660.x)
64. Elmhagen B, Angerbjörn A. 2001 The applicability of metapopulation theory to large mammals. *Oikos* **94**, 89–100. (doi:10.1034/j.1600-0706.2001.11316.x)
65. Reed D. 2004 Extinction risk in fragmented habitats. *Anim. Conserv. Forum* **7**, 181–191.
66. Horev A, Yosef R, Tryjanowski P, Ovadia O. 2012 Consequences of variation in male harem size to population persistence: Modeling poaching and extinction risk of Bengal tigers (*Panthera tigris*). *Biol. Conserv.* **147**, 22–31. (doi:https://doi.org/10.1016/j.biocon.2012.01.012)
67. Trask AE, Bignal EM, McCracken DI, Piertney SB, Reid JM. 2017 Estimating demographic contributions to effective population size in an age-structured wild population experiencing environmental and demographic stochasticity. *J. Anim. Ecol.* **86**, 1082–1093. (doi:https://doi.org/10.1111/1365-2656.12703)
68. Robert A, Sarrazin F, Couvet D. 2003 Variation among Individuals, Demographic Stochasticity, and Extinction: Response to Kendall and Fox. *Conserv. Biol.* **17**, 1166–1169.
69. Brook BW. 2000 Pessimistic and Optimistic Bias in Population Viability Analysis. *Conserv. Biol.* **14**, 564–566.
70. Yiming L, Zhongwei G, Qisen Y, Yushan W, Niemelä J. 2003 The implications of poaching for giant panda conservation. *Biol. Conserv.* **111**, 125–136. (doi:https://doi.org/10.1016/S0006-3207(02)00255-0)
71. Ferreira SM, Greaver C, Knight GA, Knight MH, Smit IPJ, Pienaar D. 2015 Disruption of Rhino Demography by Poachers May Lead to Population Declines in Kruger National Park, South Africa. *PLOS ONE* **10**, e0127783. (doi:https://doi.org/10.1371/journal.pone.0127783)